# Structural basis for specific inhibition of the highly sensitive ShHTL7 receptor

Umar Shahul Hameed[1],[†] (ID), Imran Haider[2],[†], Muhammad Jamil[2], Boubacar A Kountche[2], Xianrong Guo[3], Randa A Zarban[2], Dongjin Kim[2], Salim Al-Babili[2],* (ID) & Stefan T Arold[1],** (ID)

## Abstract

*Striga hermonthica* is a root parasitic plant that infests cereals, decimating yields, particularly in sub-Saharan Africa. For germination, *Striga* seeds require host-released strigolactones that are perceived by the family of HYPOSENSITIVE to LIGHT (ShHTL) receptors. Inhibiting seed germination would thus be a promising approach for combating *Striga*. However, there are currently no strigolactone antagonists that specifically block ShHTLs and do not bind to DWARF14, the homologous strigolactone receptor of the host. Here, we show that the octyl phenol ethoxylate Triton X-100 inhibits *S. hermonthica* seed germination without affecting host plants. High-resolution X-ray structures reveal that Triton X-100 specifically plugs the catalytic pocket of ShHTL7. ShHTL7-specific inhibition by Triton X-100 demonstrates the dominant role of this particular ShHTL receptor for *Striga* germination. Our structural analysis provides a rationale for the broad specificity and high sensitivity of ShHTL7, and reveals that strigolactones trigger structural changes in ShHTL7 that are required for downstream signaling. Our findings identify Triton and the related 2-[4-(2,4,4-trimethylpentan-2-yl) phenoxy]acetic acid as promising lead compounds for the rational design of efficient *Striga*-specific herbicides.

**Keywords** hydrolase; MAX2; seed germination inhibitor; *Striga hermonthica*; X-ray crystallography

**Subject Categories** Microbiology, Virology & Host Pathogen Interaction; Plant Biology; Structural Biology

## Introduction

The witchweed *Striga hermonthica* (*Orobanchaceae*) is a root parasitic plant that infests cereals, including rice, maize, sorghum, and pearl millet [1]. This parasitism stunts the host's growth, decimating

yields, particularly in sub-Saharan Africa. By causing around 7 billion dollars annual loses in cereals production, *Striga* has become one of the most serious biotic threats to food security [2]. Related species with different host specificity increasingly impact agriculture in the temperate areas of Europe, the Middle East and North Africa, and Asia [3].

The germination of *Striga* seeds depends on host-released strigolactones (SLs) [1]. SLs are carotenoid-derived phytohormones that shape plants in accordance with nutrient availability [4–7]. SLs consist of a butenolide ring (D-ring) linked to a second, more variable moiety via an enol ether bridge [7–9]. The SL receptor, DWARF14 (D14) in rice, is a α/β hydrolase [9–12]. Hydrolysis of SLs by D14 is mediated by a conserved Ser-His-Asp catalytic triad and results in covalent linking of the D-ring to the receptor and the release of the second moiety [9–13]. This covalent modification promotes the interaction of D14 with the downstream effector MORE AXILLARY GROWTH 2 (MAX2), which requires substantial restructuring of the SL binding pocket of D14 to fit the MAX2 binding surface [12–14] and leads to the proteasome-mediated degradation of presumed repressors of SL-inducible genes, such as the rice DWARF53 [15–17]. The protein KARRIKIN-INSENSITIVE2 (KAI2)/HYPOSENSITIVE to LIGHT (HTL)/D14-like is a close homolog to D14, which, however, binds the smoke-derived karrikins [17,18] that induce seed germination in many land plants, but not in root parasitic plants [17].

*Striga* is an obligate parasite that cannot survive without a host, to which it needs to connect via a haustorium developed shortly after germination [19]. Hence, the perception of host SLs is a crucial step in *Striga*'s life cycle, which ensures the presence of a host and determines whether imbibed seeds can develop to adult parasites. *Striga hermonthica* has a set of SL receptors, among which ShHTL7 is by far the most sensitive. These receptors evolved from extensive *KAI2* gene duplication in *Orobanchaceae* and new functionalization toward sensing SLs instead of karrikins [20–22].

*Striga* produces up to 200,000 tiny seeds per plant, which can be easily dispersed and remain viable for more than a decade [23]. The accumulation of these seeds has led to heavy soil infestation,

1  Division of Biological and Environmental Sciences and Engineering, Computational Bioscience Research Center, King Abdullah University of Science and Technology, Thuwal, Saudi Arabia
2  Division of Biological and Environmental Sciences and Engineering, The Bioactives Lab, King Abdullah University of Science and Technology, Thuwal, Saudi Arabia
3  Core Labs, King Abdullah University of Science and Technology, Thuwal, Saudi Arabia
   *Corresponding author. Tel: +966 1280 82565; E-mail: salim.babili@kaust.edu.sa
   **Corresponding author. Tel: +966 1280 82557; E-mail: stefan.arold@kaust.edu.sa
   †These authors contributed equally to this work

making *Striga* control onerous. As a wild, non-domesticated species, the maturation of *Striga* seeds is not synchronized, resulting in a seed population where only a subset is ready to germinate while others are in an inert dormant state. Moreover, most of the damage by *Striga* occurs invisibly below ground before the weed emerges from the soil. Hence, successful *Striga* control strategies must include coping with this heterogeneous seed reservoir accumulated in the soil. Suicidal germination is a possible approach, which is based on inducing seed germination by applying SL analogs in the absence of a host, thus killing *Striga* seedlings without affecting the host's SL homeostasis [24,25]. However, this method can be only used in a narrow, difficult to determine time window that allows catching the germination-ready *Striga* seeds before sowing crop seeds. *Striga* seeds that mature after treatment remain unaffected and can still decimate the crops. A very promising complementary approach is to inhibit the germination of *Striga* seeds by applying antagonists that block *Striga* SL perception [26,27]. If such inhibitors do not bind to host SL receptors, they can be applied in the presence or absence of the host as specific herbicides that enable *Striga* control throughout the growing season. However, such specific inhibitors are currently not available. Moreover, an experimental 3D protein structure of ShHTL7 that allows specific *in silico* screening for inhibitors is still missing.

# Results

### Structural basis of Triton binding to ShHTL7

We obtained 1.2–1.7-Å resolution X-ray diffraction data from crystals of ShHTL7 and the inactive active site mutant ShHTL7$_{S95C}$. Both the wild-type and mutant proteins were crystalized either without a ligand or soaked with the SL analog GR24 (Table 1). We determined their structures by molecular replacement, using the ShHTL5 structure as a template [21]. Surprisingly, in all crystals, ShHTL7 associated with the 4-(1,1,3,3-tetramethylbutyl)-phenyl hydrocarbon moiety of Triton X-100 that we used during protein purification (Figs 1A and B and EV1A). Isomorphous crystal structures of ShHTL7 purified without using Triton exhibited empty binding pockets (Fig EV1B), confirming the source and identity of this compound.

Triton bound to the deep active site pocket, formed by a double V-shaped arrangement of helices α4–α5 and α6–α7. The hydrocarbon moiety of Triton established hydrophobic interactions with Y26 (loop following β2), L124, F134 (β5–α4 loop), M139, T142, L143, L146 (on α4), F150 (α4–α5 loop), L153, T157, L161 (α5), T190, I193, C194 (α7), and M219 (β6–α8 loop; Figs 1C–E and EV2). These residues also constitute the SL binding site, as inferred by homology to rice D14 [11]. However, Triton remained at a distance of more than 8 Å from the catalytic serine S95 in the active site (Fig 1C). In ShHTL7$_{S95C}$, the sulfhydryl group in C95 was oxidized to a sulfinic acid, as confirmed by mass spectrometry (Fig EV3), causing only a minor rearrangement of the H94 side chain in otherwise identical structures (RMSD of 0.108 Å; Appendix Fig S1). Well-ordered water molecules filled the space between Triton and the bottom of the active site pocket (Fig 1C). These water molecules were well preserved in crystals of ShHTL7 purified without Triton, showing that Triton binding did not displace or rearrange this pre-existing water network.

The Triton polyethylene head group protruded out of the active site pocket (Fig 1A–D). Five ethylene oxide units were visible in a relatively weak and poorly defined electron density, suggesting weak association of this moiety. The headgroup-ShHTL7 interactions with the outer rim of the ShHTL7 binding pocket comprised a water-mediated hydrogen bond (from the oxide of the first unit to the backbone nitrogen of E135) and hydrophobic contacts (between carbons of the first unit and a hydrophobic pocket formed by ShHTL7 V138, M139, and T142, and between carbons of the fourth unit and L160 and I218). This orientation might be influenced by crystal contacts, which occlude part of the accessible protein surface.

### ShHTL7 binds strongly and specifically to Triton *in vitro*

A time course of intrinsic Triton fluorescence assays revealed a very slow association for both, ShHTL7 and the inactive ShHTL7$_{S95C}$, which converged only after 2.5 h of incubation at dissociation constants ($K_d$) of $284 \pm 22$ nM and $195 \pm 21$ nM, respectively (Fig 2A and B). These experiments were performed in a standard buffer containing 0.005% of the detergent Tween-20. In the absence of Tween-20, ShHTL7 bound to Triton with a final $K_d$ of $233 \pm 30$ nM, showing that the presence of Tween-20 did not significantly lower the affinity (Fig EV4A). The fact that the Triton-ShHTL7 complex withstood size exclusion chromatography and extensive dialysis corroborated the high affinity of the Triton association, despite the apparent slow on-rate. ShHTL7 also exhibited aggregation temperature ($T_{agg}$) increases in a Triton concentration-dependent manner after 2.5 h of pre-incubation (Fig 2C). Conversely, n-dodecyl-β-D-maltoside (nDM, a structurally different detergent with a critical micellar concentration similar to that of Triton) did not increase the protein $T_{agg}$ under the same conditions (Fig EV4B). To further quantitate the binding, we used microscale thermophoresis measurements (MST) after 2.5 h of Triton pre-incubation, which resulted in a dissociation constant ($K_d$) of $0.44 \pm 0.05$ μM (Fig 2D). We carried out a drug affinity response target stability (DARTS) assay to establish whether Triton could stabilize ShHTL7 in the presence of proteases, similarly to GR24 [26]. Both Triton and GR24 efficiently protected the protein from being digested by proteinase K, whereas point mutants that specifically lack binding to either GR24 or Triton failed to protect ShHTL7 from digestion (Fig EV4C).

### Triton efficiently blocks binding of SLs to ShHTL7

ShHTL7 bound Triton with a slightly stronger affinity than GR24 (apparent $K_d$ of GR24 was $0.92 \pm 0.01$ μM in MST, and $0.39 \pm 0.05$ μM in tryptophan fluorescence assays; Fig 2E and F). To test whether Triton inhibits SL binding, we performed a competition assay using Yoshimulactone Green (YLG) as SL ligand. In the absence of Triton, 3 μM of ShHTL7 hydrolyzes 1 μM of YLG within 50 min (Appendix Fig S2A). Pre-incubation with Triton for 2.5 h blocked ShHTL7 hydrolysis activity with an IC$_{50}$ of $0.47 \pm 0.11$ μM, which is about half the IC$_{50}$ of GR24 ($0.87 \pm 0.06$ μM; Fig 2G and H), and 25-fold lower than that reported for the non-specific inhibitor Soporidine [26]. The IC$_{50}$ value for GR24 obtained in our YLG assay is comparable to previously published YLG assays values [21]. Interestingly, the EC$_{50}$ of GR24 for ShHTL7 in transgenic *Arabidopsis* is in the picomolar range [22], suggesting that unknown

**Table 1.** X-ray diffraction data collection and refinement statistics for ShHTL7.

| Crystal | ShHTL7 (Apo) | ShHTL7-Triton | ShHTL7$_{S95C}$-Triton | ShHTL7 (Apo) |
|---|---|---|---|---|
| Resolution (Å) | 35.09–1.69 (1.75–1.69) | 40.09–1.20 (1.24–1.20) | 19.45–1.42 (1.47–1.42) | 40.09–1.35 (1.39–1.35) |
| Space group | I 222 | P 65 | P 65 | P 65 |
| Cell dimensions, *a, b, c* (Å) | 73.98, 84.04, 90.55 | 92.59, 92.59, 80.26 | 92.57, 92.57, 80.19 | 92.59, 92.59, 80.36 |
| No. of unique reflections | 31,488 (2,925) | 121,689 (12,066) | 72,852 (7,139) | 85,478 (8,298) |
| Redundancy | 9.8 (7.6) | 9.9 (9.6) | 14.4 (13.8) | 9.1 (4.5) |
| Completeness (%) | 99.23 (93.00) | 99.61 (98.67) | 99.72 (97.66) | 99.61 (96.83) |
| Mean I/σ (I) | 11.99 (0.98) | 24.56 (1.87) | 30.40 (4.52) | 23.12 (1.27) |
| R-merge | 0.103 (1.49) | 0.043 (1.07) | 0.056 (0.76) | 0.04497 (1.01) |
| R-pim | 0.034 (0.57) | 0.014 (0.36) | 0.015 (0.21) | 0.015 (0.53) |
| CC$_{1/2}$ | 0.998 (0.45) | 1.000 (0.72) | 1.000 (0.88) | 0.999 (0.58) |
| Refinement | | | | |
| Resolution (Å) | 35.09–1.69 (1.75–1.69) | 40.09–1.20 (1.24–1.20) | 19.45–1.42 (1.47–1.42) | 40.09–1.35 (1.39–1.35) |
| No. of reflections: working, test | 31,485 (2,924), 1,575 (146) | 121,220 (11,907), 6,061 (593) | 72,847 (7,138), 3,642 (356) | 85,379 (8,256), 4,269 (415) |
| R-work | 0.194 | 0.124 | 0.101 | 0.143 |
| R-free | 0.220 | 0.137 | 0.120 | 0.158 |
| No. of atoms in Protein | 2,088 | 2,552 | 2,226 | 2,183 |
| No. of atoms in ligands | 6 | 47 | 47 | 0 |
| No. of atoms in solvent | 212 | 289 | 286 | 270 |
| Protein residues | 268 | 268 | 268 | 268 |
| RMS (bonds) (Å) | 0.010 | 0.033 | 0.016 | 0.009 |
| RMS (angles) (°) | 0.87 | 1.79 | 1.69 | 1.38 |
| Ramachandran favored (%) | 98.12 | 98.11 | 99.24 | 99.25 |
| Ramachandran allowed (%) | 1.88 | 1.52 | 0.76 | 0.75 |
| Ramachandran outliers (%) | 0.00 | 0.38 | 0.00 | 0.00 |
| Rotamer outliers (%) | 1.69 | 2.02 | 0.00 | 0.00 |
| Clash score | 6.00 | 5.17 | 1.56 | 1.83 |
| Average B-factor (Å$^2$) | 42.63 | 17.49 | 18.07 | 21.65 |
| Protein | 41.85 | 15.44 | 15.82 | 19.60 |
| Ligands | 67.84 | 41.33 | 35.02 | 0.00 |
| Solvent | 49.60 | 31.79 | 32.74 | 38.23 |

Values in parentheses are for the highest resolution shell.

additional factors may play a role in this *in vivo* assay (e.g., local enrichment or co-factors).

**Triton blocks structural changes and downstream signaling of ShHTL7 induced by GR24**

Triton-free ShHTL7 structures obtained in the same space group (P65) as the Triton-bound structures did not exhibit significant changes in the binding site (Appendix Fig S2B). However, the P65 crystal packing implicated helices α4 and α5 of the binding pocket and may have stabilized the Triton-binding conformation even in the apostate. Indeed, apo-ShHTL7 obtained from a different crystal form (I222) where α4 is not constrained, showed that α4 had moved inward by about 2 Å, as compared to the P65 structures (Fig 2I). In this conformation, a steric overlap with α4 residue T142 precludes

Triton binding, strongly suggesting that the rate-limiting factor for Triton binding is the opening of α4. In solution nuclear magnetic resonance (NMR) spectroscopy, the changes in position and/or intensity of several well-dispersed chemical shifts between apo [15]N-labeled ShHTL7 and [15]N-ShHTL7 purified in the presence of Triton were of the amount and magnitude expected from our crystallographic analysis (Figs 3A and EV5). This also unequivocally confirms the direct residue-specific interaction between Triton and ShHTL7 in solution, without any major conformational changes from the protein, in line with our crystallographic analysis. Small-angle X-ray scattering (SAXS) data of apo-ShHTL7 and ShHTL7 incubated with Triton were highly similar (Appendix Fig S3A and B; radius of gyration, $R_g$, were 19.7 ± 0.2 Å and 19.5 ± 0.2 Å for apo and Triton-bound forms, respectively) and fitted the experimental structures very well (Appendix Fig S3A and B; $\chi^2$ were 2.46 and

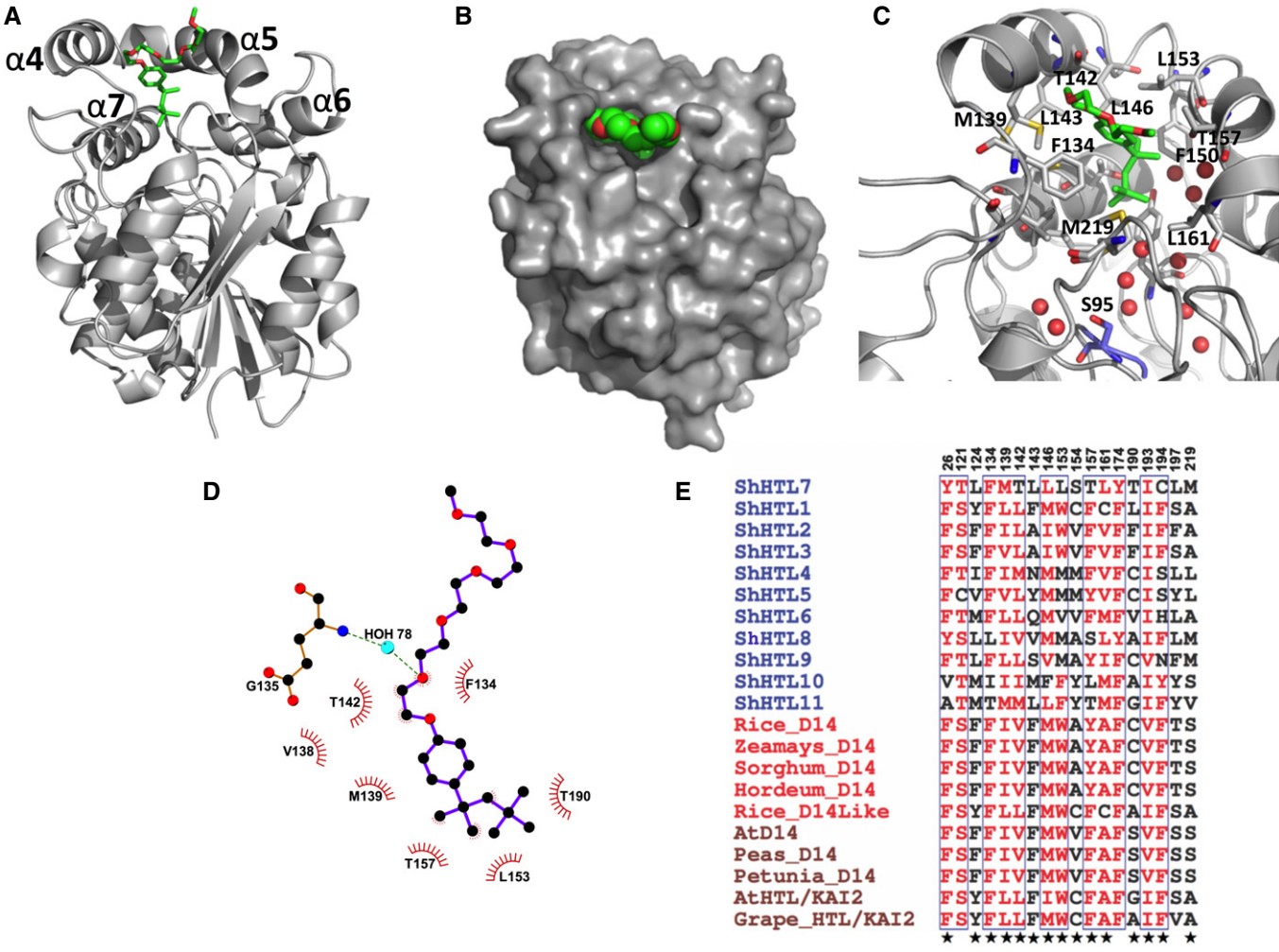

**Figure 1. Mechanism of Triton binding to ShHTL7.**

A  Crystal structure of ShHTL7 (gray) bound to Triton (stick model; green: carbons; red: oxygens).

B  Surface (protein) and sphere (Triton) representation of (A) tilted by ~10° toward the viewer.

C  Zoom into the ShHTL7 binding pocket. Key side chains and Triton are shown as stick models. Red spheres depict ordered water molecules.

D  LigPlot of the ShHTL7-Triton interactions. Hydrophobic interactions: red arc with spokes radiating toward the ligand atoms involved; hydrogen bonds: dashed lines.

E  Multiple sequence alignment of the ShHTL7 SL binding site against homologs from various species. Red residues: similar stereochemical character. Boxed: highly conserved positions. Residues marked with a star contribute to Triton interactions. For a full alignment, see Fig EV2.

3.8 Å for apo and Triton-bound ShHTL7, respectively). Collectively, these analyses validated our crystallographic analysis in solution and ruled out major structural changes upon Triton-binding.

Conversely, the titration of $^{15}$N-ShHTL7 with GR24 resulted in much more pronounced chemical shift changes (Fig 3B). Pre-incubation of $^{15}$N-ShHTL7 with Triton before adding GR24 precluded these chemical shift changes, supporting that Triton can effectively block GR24-induced structural changes in ShHTL7 (Fig 3C and Appendix Fig S3C). Accordingly, pre-incubation of ShHTL7 with Triton efficiently blocked the $T_{agg}$ changes in ShHTL7 that were otherwise introduced by GR24 in a concentration-dependent manner (Fig 3D). Of note, GR24 binding destabilized ShHTL7 ($T_{agg}$ changes in the presence of GR24 are negative), whereas Triton binding stabilized ShHTL7 (Fig 3D).

Structural analysis of the *Arabidopsis thaliana* GR24-D14 in a complex with the downstream activator AtMAX2 showed that a rearrangement of D14 helices α5 and α6, resulting in the collapse of the ligand-binding pocket, is necessary for an association with MAX2 [11]. Homology modeling of the ShHTL7:ShMAX2 complex based on the *A. thaliana* D14:MAX2 structure showed that the interface is conserved (13 residues are identical, one is homologous in ShMAX2, and all 14 residues are identical in ShHTL7; ShMAX2 and AtMAX2 share 79% sequence identity in the binding site; Appendix Fig S4), suggesting that ShHTL7 requires the same type of structural rearrangements as D14 to bind to its cognate MAX2 molecule (Appendix Fig S5A–C). SAXS data recorded on ShHTL7 incubated with GR24 showed indeed a decrease in the characteristic SAXS peak between $q = 0.2$ and 0.35 Å−1, as expected from a collapse of the large active site pocket in the ShMAX2-bound form of ShHTL7 (Appendix Fig S3C). Our observation that GR24-binding decreased the $T_{agg}$ of ShHTL7 further supported that GR24 destabilizes the ShHTL7 structure. Also, our NMR analysis of GR24-bound ShHTL7 was compatible with a loss of helix content in the SL-bound form and showed that this loss of

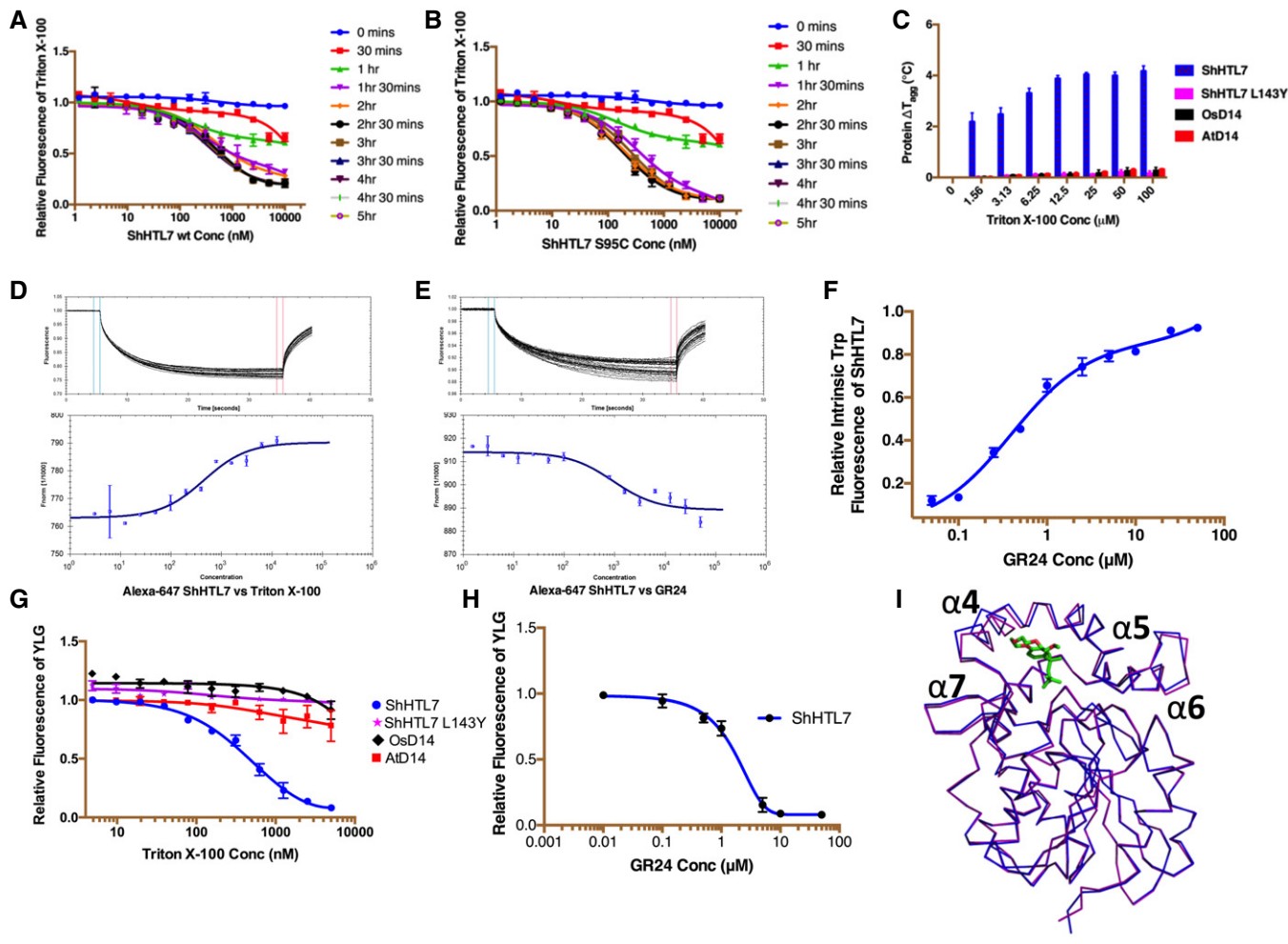

**Figure 2.  *In vitro* evaluation of Triton binding.**

A, B    Relative intrinsic Triton fluorescence for different incubation times, plotted against wild-type ShHTL7 concentrations (A) and ShHTL7 S95C mutant concentrations (B). Data are mean ± SD, $n = 3$.

C    The change in protein aggregation temperature ($\Delta T_{agg}$) plotted against Triton concentrations. Data are mean ± SD, $n = 3$.

D, E    Thermophoresis data of fluorescently labeled ShHTL7 binding to different concentrations of Triton (D) and GR24 (E) from 25 μM to sub-nanomolar range. Data are mean ± SD of the fluorescence counts from the labeled protein, $n = 3$.

F    Relative intrinsic tryptophan fluorescence of ShHTL7 is plotted against GR24 concentrations.

G, H    YLG hydrolysis by ShHTL7, ShHTL7$_{L143Y}$, and D14 proteins in the presence of increasing Triton (G) and GR24 (H) concentrations. Data are mean ± SD, $n = 3$.

I    Apo-ShHTL7 from the I222 space group (blue), superimposed onto the Triton-bound ShHTL7 crystallized in P65 (purple). Inward movement of α4 in I222 precludes binding of Triton (green) with r.m.s.d. of 0.25 Å.

helix content was blocked by Triton (Appendix Fig S6). Collectively, our analysis suggested that Triton can block the association of ShHTL7 with MAX2 even in the presence of GR24 by precluding the structural rearrangement of ShHTL7 necessary for binding to ShMAX2 (Fig 4A and Appendix Fig S6). Accordingly, pre-incubation of ShHTL7 with Triton inhibited the GR24-dependent interaction of ShHTL7 with the *S. hermonthica* downstream activator ShMAX2 (Fig 4B and Appendix Fig S5C and S6).

**Triton blocks germination of *Striga* through ShHTL7, but does not affect rice plants**

Next, we exposed different *Striga* seed batches to increasing concentrations of Triton and GR24. We observed a ~ 50%

reduction in GR24-induced germination in the presence of 15 μM Triton. This result is not explained by a general inhibitory effect of detergent on *Striga* seeds, because under the same conditions, the application of the detergent n-dodecyl-β-D-maltoside did not affect germination (Fig 4C). Three-dimensional homology modeling of other *Striga* SL receptors (ShHTL1-11) indicated specificity of Triton for ShHTL7 (Appendix Text S1). The active site composition of other ShHTLs, especially ShHTL5 and 6, appeared to preclude Triton binding because of steric clashes (in ShHTL6: T142L, T157F, and C194H; in ShHTL5: T142L and T157Y; Appendix Fig S7). Thus, the observed effect is expected to be solely caused by the inhibition of ShHTL7.

For field application, *Striga* seed germination inhibitors should not bind to ShHTL homologs in non-parasitic plants. We tested

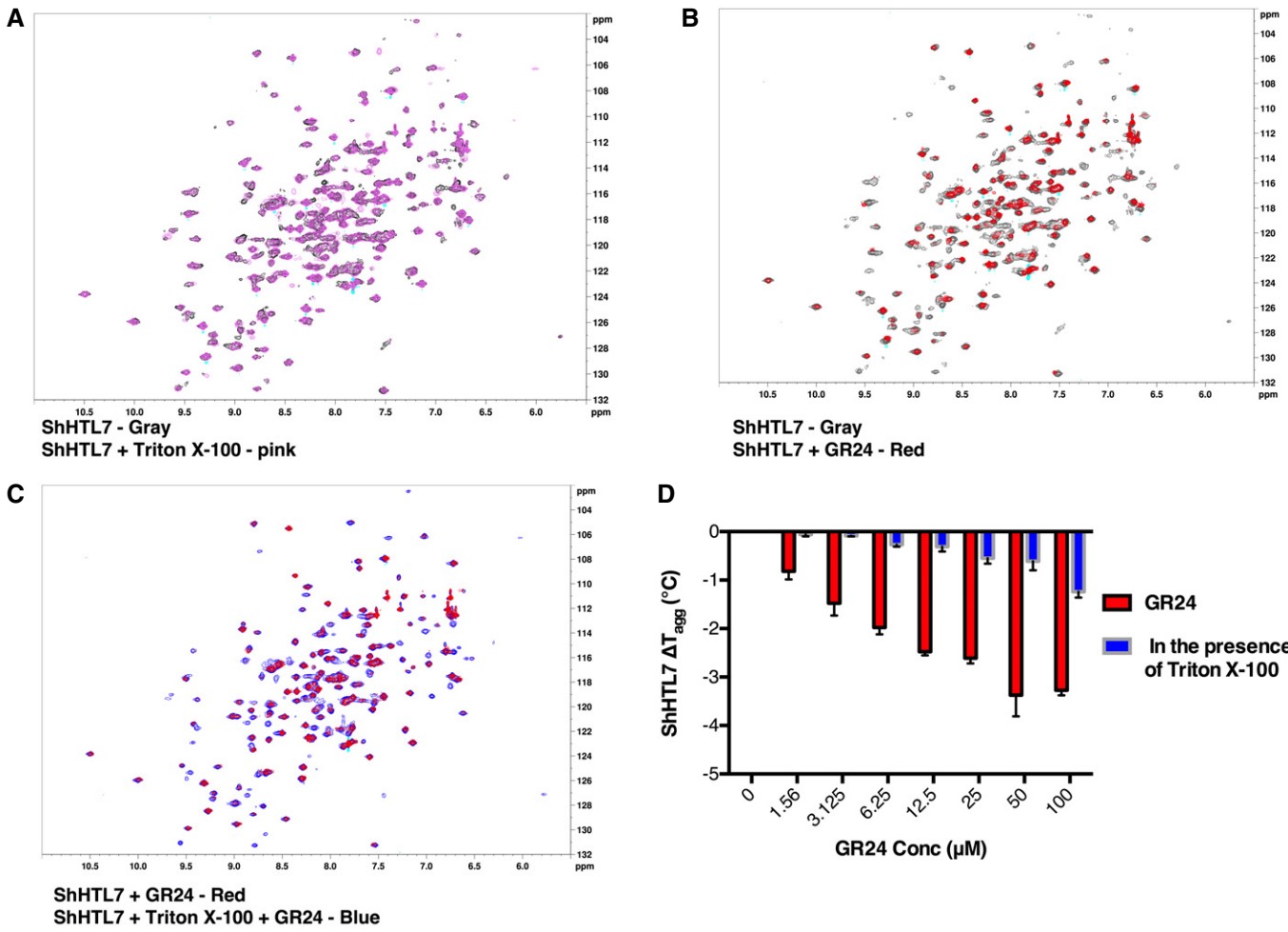

**Figure 3.  Triton binds to ShHTL7 and stabilizes the protein from structural changes.**

A    $^{15}$N HSQC spectra of 0.3 mM ShHTL7 (gray) overlapped on ShHTL7 purified in the presence of Triton (pink).

B    $^{15}$N HSQC spectra of 0.3 mM ShHTL7 (gray) with spectra colored red at the final ShHTL7–GR24 ratio of 1:1.5.

C    $^{15}$N HSQC spectra of 0.3 mM ShHTL7 + Triton (dark blue) overlapped on ShHTL7 (red) at the final ShHTL7–GR24 ratio of 1:1.5.

D    The change in ShHTL7 (red) and ShHTL7 + Triton (blue) aggregation temperature ($\Delta T_{agg}$) plotted against GR24 concentrations. Data are mean ± SD, $n = 3$.

*in vitro* binding of Triton to *Arabidopsis* and rice D14 (AtD14 and OsD14, respectively). Triton fluorescence did not change in the presence of up to 5 μM of AtD14 or OsD14 (Fig 4D), and the presence of up to 100 μM of Triton failed to affect the $T_{agg}$ of AtD14 and OsD14, ruling out strong association with these proteins (Fig 2C). Consistent with these results, application of Triton at concentrations up to 0.5 μM did not impact OsD14 or AtD14 YLG hydrolysis activity. Even at a 5 μM concentration, Triton decreased the YLG hydrolysis rate of OsD14 or AtD14 by only 10% (Fig 2G). Importantly, application of 15 μM of Triton, a concentration that inhibited *S. hermonthica* germination by 50%, did neither inhibit rice seed germination nor affected seedling growth (Fig 4E and Appendix Fig S8).

Structural analysis provided a rationale for the absence of D14 binding. The ligand specificity of HTL/D14/KAI2 proteins is determined by the positioning of helices α4 and α5, which frame the binding pocket entrance, and by the residues that line the binding pocket (Figs 5A and B, and EV2). Although most secondary structure elements of ShHTL7 superimpose closely on those of AtD14 and OsD14 (RMSD of 1.12 Å), the α4–α5 aperture of ShHTL7 is widened by ~ 1.5 Å in its I222 apo form and by ~ 3 Å in its Triton-bound form through an outward movement of α4 (Fig 5A and B). This α4 opening appears precluded in non-*Striga* orthologues by the substitution of ShHTL7 L143 with phenylalanine in the α4–α7 interface (Figs 1C and E, and EV2). This substitution is homologous and stereochemically equivalent to the L143Y substitution seen in ShHTL5, for which the crystal structure also shows a more inward positioning of α4 (Figs 1E and 5A and C). In agreement, Triton failed to bind the ShHTL7 L143Y mutant *in vitro* and did not inhibit its hydrolysis activity or ShMAX2 binding (Figs 2G and 4B and D). Additionally, the substitution of several side chains in the ShHTL7 binding pocket with bulkier ones in non-*Striga* D14/KAI2 (L153W, T157F/Y, C194F; Figs 1E and 5B) reduces the size of the binding pocket more than is tolerable by Triton (Fig 5B). Conservation of the active site in D14/KAI2 proteins (Figs 1E and EV2) supports generalization of these results.

   

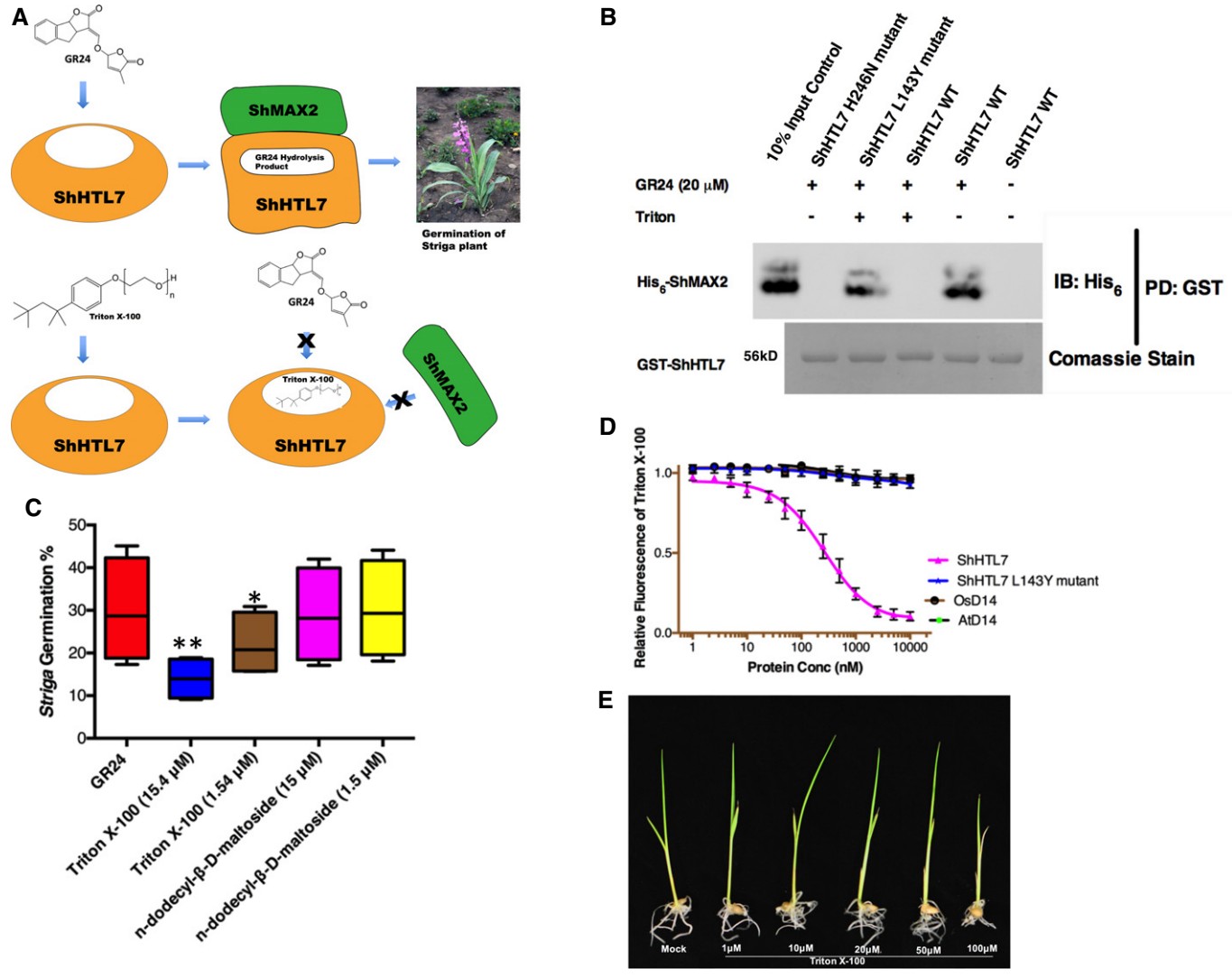

**Figure 4.  *In vitro* and *in vivo* efficacy of Triton in combating activity of ShHTL7 and *Striga*.**

A   Schematic diagram depicting the mode of inhibition of ShHTL7 by Triton via blocking the binding pocket of ShHTL7 and thus prevents the access for its substrate Strigolactones such as GR24.

B   GST pulldown of His$_6$-ShMAX2 with GST-ShHTL7 in the presence of Triton and GR24. The ShHTL7 mutations L143Y and H246N block binding to Triton and GR24, respectively. Ten percent of used ShMAX2 are shown as input control. Full gels are shown in Appendix Fig S5C.

C   Percentage of *Striga hermonthica* seeds germinating after 2 days of treatment with Triton at 15.4 (0.001%) and 1.54 μM (0.0001%) concentrations and n-dodecyl-β-D-maltoside at 15 and 1.5 μM concentrations in the presence of GR24 at 1, 0.5, 0.25, and 0.125 nM concentrations, respectively. Data are shown as box and whisker plots, where horizontal central value represents median, upper and lower quartiles are the ends of the box, the box spans the interquartile range and whiskers are the maximum and minimum values, $n = 6$ (50–100 seeds per replicate), *$P < 0.05$, **$P < 0.01$ (one-way ANOVA).

D   Intrinsic Triton fluorescence in the presence of increasing concentrations of ShHTL7 and homologs. Data are mean ± SD, $n = 3$.

E   Triton effects on Nipponbare rice seedling growth. Representative 2-week-old Nipponbare rice seedlings treated with Triton or water (mock).

## Triton-like structures as lead compounds for specific inhibition of *Striga* seed germination

Our crystal structures showed that the Triton polyethylene oxide chain also contributes to the binding to ShHTL7. To assess the impact of this chain, we tested commercially available Triton-like compounds with ethylene oxide group sizes from 1 to 40 units (Triton X-100 has 9–10 units; Fig 6A–F). We also included 2-[4-(2,4,4-trimethylpentan-2-yl)phenoxy]acetic acid (PAA). Although it has the same hydrophobic tail as Triton, PAA is not a detergent because it

carries a phenoxy acid group instead of the polyethylene repeats. All Triton-like compounds inhibited YLG hydrolysis with IC$_{50}$ values of 1–12 μM that were higher, but still in the same range as the one for Triton (0.47 μM; Fig 6G). PAA showed the strongest inhibitory effect on ShHTL7 activity in YLG hydrolysis assays (IC$_{50}$ of 1.48 μM; Fig 6E and G). Moreover, PAA showed direct binding to ShHTL7 in MST experiments, with a $K_d$ of 1.7 ± 0.24 μM, and stabilized ShHTL7 $T_{agg}$ with the addition of PAA (Fig 7A and B). 10 μM PAA also inhibited *S. hermonthica* germination at ratios up to 40% (Fig 7C and Appendix Fig S8), further establishing that the inhibitory

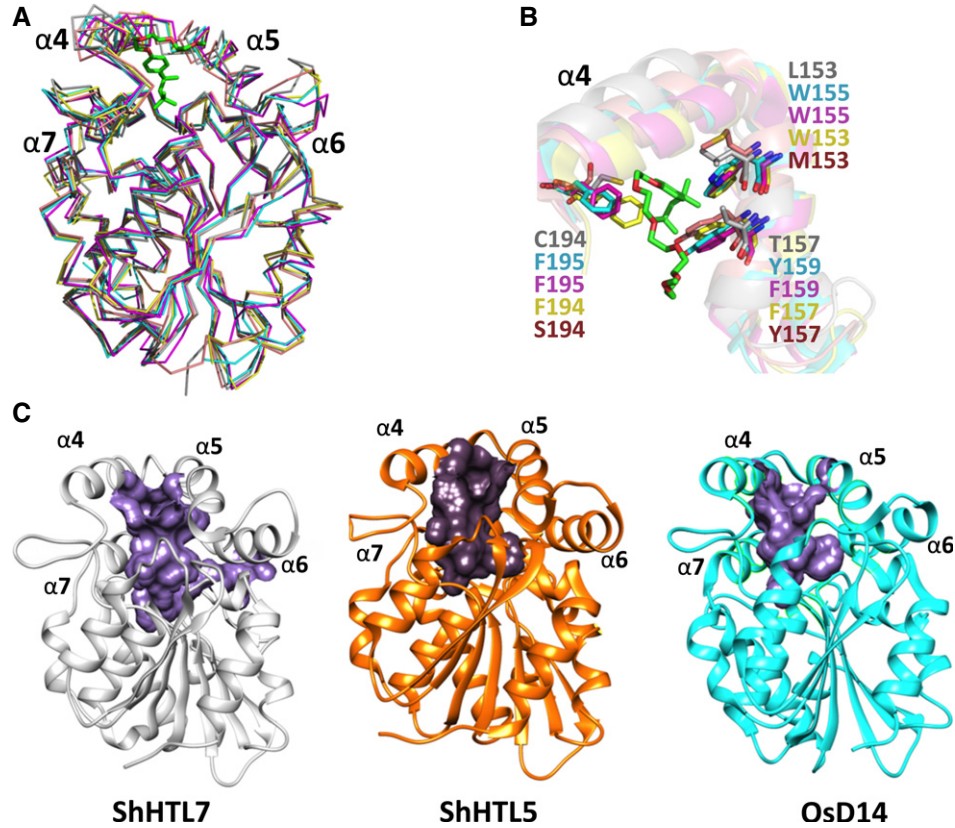

**Figure 5. Structural determinants of Triton specificity.**

A  Superimposition of ShHTL7-Triton (space group P65; gray; Triton: green carbons and red oxygens), ShHTL5 (PDB 5CBK; salmon), OsD14 (PDB 4IHA; cyan), AtD14 (PDB 4IH4; magenta) and AtHTL (PDB 4HRY; yellow). The RMSD between structures is 1.12 Å.

B  Zoom onto helices α4 and α5. Triton and relevant side chains are shown.

C  ShHTL7 (gray), ShHTL5 (orange), and OsD14 (cyan) with their ligand-binding pockets shown in surface view.

activity of Triton on SL hydrolysis and *Striga* seed germination is not simply a non-specific detergent effect. These results showed that very different head groups can achieve similar inhibitory effects *in vitro* and *in planta*, suggesting that a larger screen of this moiety may result in compounds with improved efficacy and bioavailability.

# Discussion

The witchweed has become a major threat for food security in sub-Saharan Africa, and an efficacious herbicide to control this pandemic plant pest is still lacking. Collectively, our data support that Triton-like compounds are a very promising basis for the development of specific inhibitors of *Striga* germination.

Among ShHTL family members, ShHTL7 stands out as the receptor that binds *in vitro* to all SLs tested with high affinity [21]. Our analysis now suggests that the broad specificity of ShHTL7 toward SLs is explained by the large volume of the SL binding pocket (1,174.3 Å$^3$ compared to 894.5 and 710.1 Å$^3$ in ShHTL5 and rice D14, respectively) that can accommodate diverse SLs without steric clashes (Figs 5C and 7D). The increased pocket size is caused by two mechanisms: Firstly, the increased opening between helices 4 and 5, and secondly the replacement of bulky

aromatic hydrophobic side chains (F, W, Y) with smaller ones (L, V, S, T). This second mechanism is already apparent when comparing D14 with ShHTL5; however, it is further reinforced in ShHTL7, in particular through the presence of a threonine in position 157 instead of a tyrosine (Figs 1E and 7D). Compared to ShHTL5, ShHTL7 has also substituted four of the flexible methionine residues with more rigid amino acids (L and S; Fig 1E). The resulting reduction in flexibility inside the pocket is expected to reduce the entropic penalty of binding to SLs and hence to generally increase SL affinity. Using transgenic *Arabidopsis* lines, it was previously shown that ShHTL7 is the most sensitive SL receptor in *Striga* [22]. Our observation that the ShHTL7-specific inhibition by Triton reduced germination by up to 50% provides the first direct evidence for the primary role of this receptor in *Striga* seed germination.

The site used by MAX2 (D3) to bind to AtD14 is 79% identical in ShMAX2 (Appendix Fig S4), suggesting that SL-bound ShHTLs also need to undergo the same structural rearrangement of helices α4, α5, and α6 for ShMAX2 association as AtD14 in its complex with AtMAX2. Our structural and biophysical analyses demonstrate that GR24 binding leads to loss of helical content, destabilization, and structural changes in ShHTL7, as needed to associate with ShMAX2. In both of our ShHTL7 crystal lattices, α4 and α5 are more distant

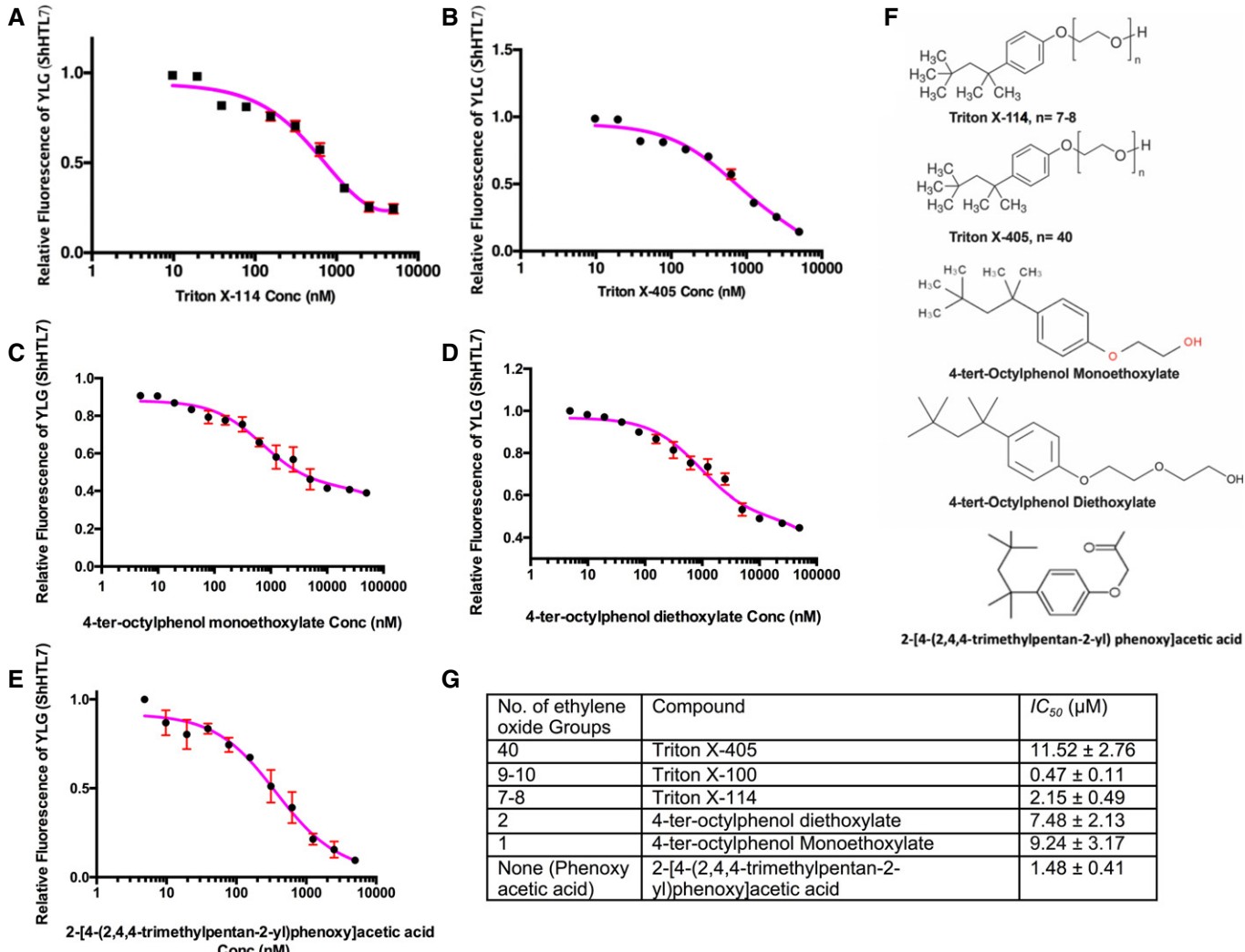

**Figure 6.  Hydrolysis activity of ShHTL7 in the presence of Triton X-100-like compounds.**

A–E  Triton X-114 (A), Triton X-405 (B), 4-ter-octylphenol monoethoxylate (C), 4-ter-octylphenol diethoxylate (D), 2-[4-(2,4,4-trimethylpentan-2-yl)phenoxy]acetic acid (E). Data are mean ± SD, $n$ = 3.

F  Chemical structures of the tested compounds.

G  $IC_{50}$ values for Triton-like compounds based on ShHTL7 YLG hydrolysis assays.

from the underlying helices α7 and α6 than in D14 or ShHTL5. α4 and α5 are also anchored with smaller side chains (Fig 7E and F). Combined, the loser attachment and shorter side chains of this region may facilitate its restructuring upon binding to ShMAX2, which may contribute to the high capacity of ShHTL7 to trigger downstream signaling.

Interestingly, the picomolar sensitivity of ShHTLs *in planta* does not correlate with their affinity for SLs *in vitro*, inferring that other factors contribute [21,22]. The discrepancy between *in vitro* and *in planta* measurements is in line with previous observations. Toh *et al* reported that picomolar concentrations of GR24 are sufficient to trigger seed germination in transgenic *Arabidopsis* lines expressing *ShHTL7*, whereas the *in vitro* YLG assay with recombinant ShHTL6 and ShHTL7 revealed an $IC_{50}$ of ~0.5 μM for GR24 [21,22]. Thus, *in vitro* binding or hydrolysis assays, GR24-based seed germination activity in transgenic *Arabidopsis*, and *Striga* germination

through natural SLs in the field cannot be compared. The apparent SL sensitivity in planta may be strongly influenced by processes such as bioavailability, local SL concentration, or interactions with other proteins. Hence, the inhibitory effect of Triton *in planta* might be limited by its bioavailability and ability to be uptaken and transported within *Striga* seeds.

Chemical control of *Striga* is currently limited due to the unavailability of compounds that specifically target *Striga*, without affecting the host plants. Here, we identified Triton X-100 as a specific inhibitor of the most sensitive *Striga* SL receptor, ShHTL7. The structural match, the high *in vitro* affinity, and the robust *in vivo* activity identify Triton as a promising lead compound for the design of *Striga*-specific herbicides. We showed that Triton X-100 derived structures with different head groups also inhibited ShHTL7, opening up possibilities of designing a new class of *Striga* inhibitors. Moreover, the Triton-ShHTL7 complex structure suggests that

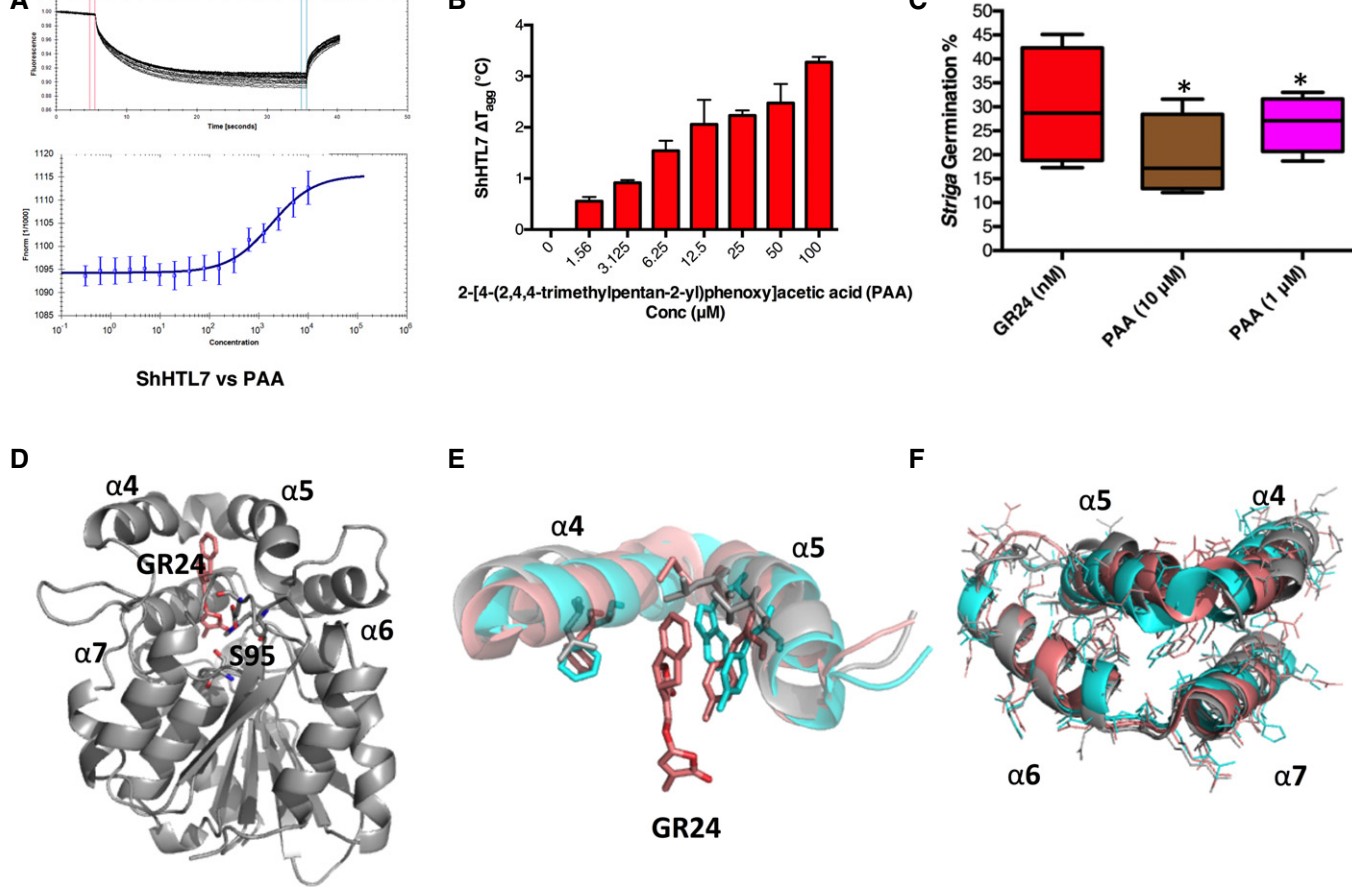

**Figure 7.   PAA binds to ShHTL7 and inhibits *Striga* germination and the structural requirement for ShHTL7 to accommodate its substrate.**

A   Thermophoresis data of fluorescently labeled ShHTL7 binding to different concentrations of PAA from 25 μM to sub-nanomolar range. Data are mean ± SD of the fluorescence counts from the labeled protein, *n* = 3.

B   The change in ShHTL7 aggregation temperature ($\Delta T_{agg}$) plotted against PAA concentrations. Data are mean ± SD, *n* = 3.

C   Percentage of *Striga hermonthica* seeds germinating after 2 days of treatment with 2-[4-(2,4,4-trimethylpentan-2-yl)phenoxy]acetic acid at 10 and 1 μM in the presence of GR24 at 1, 0.5, 0.25, and 0.125 nM concentrations, respectively. Data are shown as box and whisker plots, where horizontal central value represents median, upper and lower quartiles are the ends of the box, the box spans the interquartile range and whiskers are the maximum and minimum values, *n* = 6 (50–100 seeds per replicate), *$P < 0.05$ (one-way ANOVA).

D   ShHTL7 (space group I222; gray) superimposed with GR24 (orange stick model) transposed from its position bound to OsD14 (PDB 5DJ5).

E   Superimposition of helices α4 and α5 of ShHTL7 (gray), ShHTL5 (PDB 5CBK; salmon), OsD14 (PDB 5DJ5; cyan). The position of GR24 (bound to OsD14) and relevant side chains are shown.

F   Superimposition of helices α4, α5, α6, and α7 ShHTL7 (gray), ShHTL5 (PDB 5CBK; salmon), and OsD14 (PDB 5DJ5; cyan) and the side chains are shown as lines.

derived compounds with a hydrophobic moiety that fills the gap between Triton and the active site pocket may exhibit higher inhibitory activity. A further improvement might be achieved by designing derivatives that reach and covalently bind to the active site.

Its potency, persistence, and entire filling of the ShHTL7 active site distinguish Triton from compounds such as glycerol and PMSF, which are frequently found attached non-specifically to small regions of protein active sites. Moreover, the fact that the action of Triton can be recapitulated by the non-detergent Triton-like compound PAA, but not by the detergent nDM, excludes that our observations are simply caused by non-specific detergent effects. We show that the occupation of the ShHTL7 binding pocket by Triton precludes the structural changes required for downstream signaling through ShMAX2. The high resolution of our structural

data delivers the atomic details required for the rational development of inhibitors with improved efficacy. Indeed, our structural analysis showed that the binding pocket of ShHTL7 is too different to be modeled correctly based on known homologous structures.

Our *in vivo* assessment of *Striga* germination inhibition by Triton showed robust effects, with, however, noticeable differences between seed batches and assay conditions. Based on structural modeling of all ShHTL7 family members, the reduction of germination by Triton is predicted to be caused only by inhibition of ShHTL7. In transgenic *A. thaliana*, ShHTL4-6 showed EC50 values of 0.01–0.3 μM for GR24 [21]. These concentrations are far above the picomolar sensitivity of *Striga* seeds for SLs [28], and hence, the contributions of ShHTL family members other than ShHTL7 are expected to be of only minor importance *in planta*.

Interestingly, it was earlier noted, but not further investigated, that Triton X-100 concentrations above 0.001% hindered seed germination of *Orobanche cumana* and *Orobanche minor* [29], suggesting that this compound might also be used to combat root parasitic weeds other than *S. hermonthica*.

In conclusion, our work confirms and explains the primordial role of ShHTL7 as SL receptor in *Striga*. Our analysis paves the way for realizing a promising approach in combating *Striga*, which is based on applying *Striga*-specific germination inhibitors in the presence of host crops throughout the planting season. Thus, such Triton-derived inhibitors can complement or replace the suicidal germination approach and provide a highly needed addition to the insufficient tool-box of anti-*Striga* measures.

# Materials and Methods

## Cloning, expression and protein purification

*Striga hermonthica* ShHTL7 cDNA (GenBank accession: KR013127) was kindly provided by Prof. Tadao Asami (The University of Tokyo, Japan). ShHTL7$_{S95C}$ mutant, *A. thaliana* D14 (*AtD14*) cDNA (GenBank accession: AY097402), and *Oryza sativa* D14 (*OsD14*) cDNA (GenBank accession: BAF11220) were synthesized and cloned into PUC57 (GenScript). ShHTL7, ShHTL7$_{S95C}$ mutant, and *OsD14* were amplified by PCR using primers [30] (Appendix Table S1). The obtained fragments were digested with *Bam*HI and *Xho*I and ligated into accordingly treated pGEX-6P-1 expression vector (GE Healthcare). *AtD14* was amplified by PCR using AtD14-F/AtD14-R primers (Appendix Table S1), and the obtained fragment was digested with *Eco*RI and *Sal*I, and ligated into *Eco*RI/*Sal*I digested pGEX-6P-1 expression vector. *ShHTL7-L143Y* mutant was generated by PCR according to [30] and using ShHTL7-L143Y-F/ShHTL7-L143Y-R primers (Appendix Table S1) designed to introduce the desired mutation in the pGEX-6P-1-*ShHTL7* plasmid. After PCR, the mixture was digested with *Dpn*I and transformed into *Escherichia coli* TOP10 cells, to generate the plasmid with the introduced mutation. All plasmids were verified by sequencing (KAUST Bioscience Core Lab). The plasmids were then transformed into *E. coli* BL21 (DE3) cells. Cells were grown in LB broth containing ampicillin (100 mg/ml) at 37°C until an OD$_{600}$ of 0.6 and for NMR $^{15}$N labeling, and the cells were grown in M9 minimal media supplemented with $^{15}$N ammonium chloride. Expression was induced by adding 150 μM isopropyl β-D-1-thiogalactopyranoside (IPTG), and cultures were incubated at 16°C overnight under shaking. Cells were harvested by centrifugation at 8,000 *g* for 10 min. Cell pellet of one liter culture was resuspended in 25 ml lysis buffer (50 mM Tris–HCl (pH 8.0), 200 mM NaCl, 3 mM DTT, 700 μM Triton X-100, or 9.6 mM Tween-20) and lysed by sonication. Cell debris was removed by centrifugation at 30,000 *g* for 30 min, and proteins were purified from the obtained supernatant using Glutathione Sepharose 4B resins (GE Healthcare). The N-terminal GST tag of ShHTL7 was removed by overnight incubation with PreScission Protease (GE Healthcare) at 4°C. After GST cleavage, protein was further purified on a HiLoad16/60 Superdex 200 prep-grade gel filtration column (GE Healthcare) using a buffer containing 10 mM HEPES (pH 7.5), 50 mM KCl, 3 mM DTT. The purified protein was concentrated to 10 mg/ml and stored at −80°C. The expression and purification protocol for ShHTL7 was also used for AtD14 and OsD14 proteins, except that AtD14 and OsD14 were eluted from glutathione resins using 0.2 mM reduced glutathione (Sigma) without cutting the GST tag. Eluted proteins were passed through gel filtration equilibrated with buffer containing 10 mM HEPES (pH 7.5), 50 mM KCl, 3 mM DTT and further used for *in vitro* assays.

## Crystallization and structure determination

The hanging drop vapor diffusion method was used for crystallization. Initial plate crystals were obtained for apo-ShHTL7 and ShHTL7 co-crystallized with 1 mM *rac*-GR24 (Chiralix) by equilibrating 1.0 μl of protein (15 mg/ml) mixed with 1.0 μl of reservoir solution (0.2 M magnesium chloride, 0.1 M Tris–HCl pH 8.5, and 30% PEG 4000). These crystals were crushed and used as microseeds to obtain diffraction-quality crystals under same crystallization conditions. For ShHTL7 and the ShHTL7$_{S95C}$ mutant with Triton X-100, diffraction-quality crystals were obtained by equilibrating 1.0 μl of protein (10 mg/ml) mixed with 1.0 μl of reservoir solution (1 M lithium sulfate, 0.1 M HEPES pH 7.5), in addition apo ShHTL7 crystallized in reservoir solution containing 0.1 M MES pH 6.5 and 1 M magnesium sulfate. All crystals grew in 1–3 days at 23°C. For data collection, 25% glycerol was added to the well solution as a cryo-protectant, and the crystals were flash-cooled in liquid nitrogen. All data for apo ShHTL7, ShHTL7-Triton X-100, and ShHTL7$_{S95C}$-Triton X-100 were collected at 100K at the beamlines Proxima 1 and Proxima 2A at the SOLEIL Synchrotron (France), using a Pilatus 6 M and EIGER 6 M detector, respectively (proposal numbers 20160098, 20161236, 20170193). The data were processed in XDS. The crystal structure of ShHTL7$_{S95C}$-Triton X-100 was determined by molecular replacement using MoRDa with the ShHTL5 structure (PDB 5CBK) as search model. The refined ShHTL7$_{S95C}$ model was then used as MR template for ShHTL7-Triton X-100 and apo ShHTL7. The structures were manually inspected using Coot and refined using Phenix Refine (Table 1). The figures were drawn using PYMOL. The active site pocket volume was calculated following the previously published method [22].

## Yoshimulactone green (YLG) hydrolysis assays

*In vitro* YLG hydrolysis assays were performed using 3 μM of ShHTL7, ShHTL7-L143Y, AtD14, and OsD14 in a reaction buffer (1× PBS) with 0.1% dimethyl sulfoxide (DMSO) in a 100 μl volume on a 96-well black plate (Greiner). The fluorescent intensity was measured by SpectraMax i3 (Molecular Devices) using an excitation wavelength of 480 nm and a detection wavelength of 520 nm. ShHTL7, ShHTL7-L143Y, AtD14, and OsD14 were mixed with Triton X-100, using serial dilutions to cover the Triton X-100 concentration range from 5 μM until 4.87 nM. Protein-Triton mixes were pre-incubated for 30 min. 1 μM YLG (Tokyo Chemical Industry Co. Ltd.) was then added, and the assay was incubated for 60 min. The change in fluorescence observed over the course of 1 h incubation of YLG in buffer without protein was subtracted from the data collected in the presence of protein. The resulting relative fluorescence values were analyzed. The inhibitory curves and IC$_{50}$ values were plotted using GraphPad (Prism 6) four-parameter logistic curve. The rate of YLG hydrolysis by ShHTL7 was recorded following the above method at 10-min interval for 2 h.

## Triton X-100 fluorescence

Triton X-100 has an excitation and emission wavelength of 275 and 302 nm, respectively. ShHTL7, ShHTL7-L143Y, AtD14, and OsD14 were incubated in 50 mM Tris–HCl pH 7.5, 150 mM NaCl, with or without 0.005% Tween-20, at concentrations ranging from 10 μM to 1 nM (obtained using serial dilutions) with 154 nM of Triton X-100 for 1–2 h, and fluorescence values were then measured by SpectraMax i3 (Molecular Devices) at excitation and detection wavelengths of 275 and 302 nm, respectively. The fluorescence exhibited by Triton X-100 alone was subtracted from the fluorescence in the presence of protein. Data were normalized to the background fluorescence from the protein to obtain the change in Triton X-100 fluorescence intensity. To determine the $K_d$, the changes in Triton X-100 fluorescence intensity were plotted against the concentration of the protein using the single binding site model implemented in GraphPad (Prism 6).

## Differential static light scattering (DSLS)

The specific aggregation temperature, $T_{agg}$, at which ShHTL7, ShHTL7$_{L143Y}$, AtD14, and OsD14 aggregated in the presence or absence of Triton X-100, GR24, and PAA was measured using DSLS on the Stargazer system (Harbinger Biotechnology and Engineering Corporation, Markham, Canada). Protein samples at a concentration of 10 μM were overlaid with mineral oil in a clear bottom 384-well black plate (Corning) and heated from 20 to 95°C at 1°C/min; light scattering was detected by a CCD camera every 0.5°C in the presence or absence of varying concentrations of Triton X-100, GR24, and PAA yielding the difference in aggregation temperature at a particular concentration from apoproteins ($\Delta T_{agg}$). Data were normalized, and $\Delta T_{agg}$ plotted against the Triton X-100, GR24, and PAA concentration. Resulting data were plotted to determine the change in $T_{agg}$ of all proteins when bound to Triton X-100, GR24, and PAA using GraphPad (Prism 6).

## Mass spectrometry (MS) for molecular weight determination

Mass spectrometric analysis of ShHTL7-apo, ShHTL7$_{S95C}$, and ShHTL7 bound to Triton X-100 was performed using electrospray ionisation system Q-ToF (Bruker Maxis) equipped with liquid chromatography (Agilent Technologies). Under denaturing conditions, the protein was desalted using C4 Ziptip (EMD Millipore) and eluted into 50% acetonitrile/50% water/0.1% formic acid. For each analysis, a 3–6 μl aliquot of the protein solution was loaded. Mass spectra were processed by inbuilt software for Bruker maxis.

## *Striga hermonthica* germination bioassays

*Striga hermonthica* seeds were collected from *Sorghum bicolor* field in Sudan. For germination bioassays, *Striga* seeds were first sterilized and pre-conditioned as described before [31]. The pre-conditioned *Striga* seeds were treated with Triton X-100 concentrations 1.54 and 15.4 μM along with GR24 (final concentration of 1.0, 0.5, 0.25, and 0.125 nM). Triton X-100/GR24 mixtures were applied (50 μl per disk) in four replicates. Corresponding volumes of GR24 and sterile MiliQ water were included as positive and negative control, respectively. The petri dishes were sealed with parafilm,

enfolded with aluminum foil, and incubated for 48 h at 30°C. The germinated (seeds with radicle emerging through the seed coat) and non-germinated *Striga* seeds were recorded under a binocular, and germination percentage was calculated. For germination assays against n-dodecyl-β-D-maltoside, it was done at 15, 1.5 μM and for PAA it was 10, 1 μM, respectively, and same method as above was adopted for the assays with same GR24 concentrations.

## Rice seeds germination assays and growth of seedlings

The rice seeds *cv.* Nipponbare were surface-sterilized using 50 ml (2.5%) sodium hypochlorite with 0.4% of Tween-20 for 15 min. After six subsequent washing steps with sterilized water, the rice seeds were kept in water at 30°C for overnight imbibition. About 2 ml of Triton X-100 concentrations ranging from 1 to 100 μM and sterile MilliQ water (control) was added in 24-well plates with six replications. One sterilized rice seed was put in each well and allowed to germinate for 48 h at 30°C. The germinated (radicle emergence) and non-germinated seeds were recorded, and germination percentage was calculated. Then, same seeds were allowed to grow under light in growth cabinet for another 1 week. The length of 2-week-old seedlings was measured with roller and averaged.

## GST pulldown of ShMAX2

Purified GST-ShHTL7 wt, ShHTL7 L143Y, and ShHTL7-H246N mutants were bound to the Glutathione Sepharose 4B Beads, where Triton is used in the lysis step and in certain samples Tween-20 was used instead of Triton. Bound proteins were incubated with 20 μM GR24 and His$_6$-ShMAX2 lysate for 2 h and then washed with lysis buffer for three times and then ShHTL7 wt and mutants were eluted with the addition of 20 mM reduced glutathione. Thus, eluted protein was subjected to Western blot using antibody specific to 6XHis (anti-His).

## Ligand-binding and stabilization of ShHTL7 by chemical shift perturbation

To determine ShHTL7 ligand binding and stabilization by NMR, the chemical shift changes induced in $^{15}N$ HSQC spectra of ShHTL7 by the presence of GR24 and simultaneously in the presence of Triton X-100 was recorded. All spectra were recorded at 298 K on a Bruker Avance 950 MHz spectrometer equipped with TCI cryoprobe. Single-labeled $^{15}N$ recombinant ShHTL7 (0.3 mM) was titrated by addition 0.45 mM unlabeled GR24, and $^{15}N$-labeled ShHTL7 (0.3 mM) purified in the presence of Triton X-100 was titrated by addition 0.45 mM unlabeled GR24. The peak shifts were analyzed using Topspin software program, and the data were overlapped to identify the changes in the chemical shift.

## Micro scale thermophoresis (MST) for ShHTL7 binding to ligands

ShHTL7 was N-terminally labeled with RED dye (NT-647) following the protocol from manufacturer (NanoTemper, Germany). Free dye was removed by passing through the Sephadex G-25 column. Serial dilutions of unlabeled Triton X-100, GR24, and PAA ranging from 25 μM to sub-nanomolar range were mixed with 20 nM of

NT-647-labeled ShHTL7 in buffer (50 mM Tris–HCl, 150 mM NaCl, 0.005% Tween-20) and incubated for 2.5 h in the studies for Triton X-100 and PAA except with GR24, and it was recorded after 10-min incubation. About 10 μl of sample was loaded into premium monolith NT capillaries, and assays were carried out in a NanoTemper Monolith NT.015T. The red fluorescence was excited by a light-emitting diode, and its emission was recorded on the focused part of capillaries. Using laser diode, a microscopic temperature gradient was created at the same location and the fluorescence depletion was measured. $K_d$ values were determined by plotting the concentrations of unlabeled ligands against the changes in fluorescent thermophoresis signal, and the data were analyzed using the NanoTemper analysis software.

### Drug affinity response target stability (DARTS) assay

ShHTL7 and mutants were diluted to a concentration of 30 μM in 50 mM Tris–HCl pH 7.5, 150 mM NaCl, 0.005% Tween-20, incubated with GR24 at final concentration of 100 μM for 5 min, and separately with 100 μM Triton X-100 for 2 h. One sample was kept aside for use as an undigested control. Proteinase K (Hampton Research) was then added to ShHTL7 protein to a final protease concentration of 10 μg/ml. Reactions were incubated at room temperature for 5 min and then SDS–PAGE loading buffer was added to stop the reaction, and the samples were immediately boiled for 3 min. Equal amounts of each sample, corresponding to 10 μg of protein, were run on a 4–20% gradient polyacrylamide gel (Bio-Rad) and analyzed by Coomassie Blue staining [32].

### Small-angle X-ray scattering

Experiments were performed at the SWING beamline (SOLEIL, Saint-Aubin, France), using the Biology Laboratory 1 (proposal number 20170193). The wavelength was set to λ = 1.03 Å. The distance of the sample to the detector was 1.8 m, resulting in the momentum transfer range of $0.01\ \text{Å}^{-1} < q < 0.5\ \text{Å}^{-1}$. Between 30 and 50 μl of protein was used. For each sample, two concentrations and two buffer blanks were measured. Protein and buffer data were selected and averaged. Buffer data were subtracted from the protein data using SWING's on-site software. Programs of the ATSAS suite [33] were used for data processing, Guinier plots, distance distributions (PRIMUS), model fitting (CRYSOL), and figure preparation (SASPLOT).

### Data availability

The coordinates for the structures reported in this paper have been deposited in PDB under accession numbers 5Z8P, 5Z95, 5Z82, and 5Z89.

**Expanded View** for this article is available online.

### Acknowledgements
We acknowledge SOLEIL for provision of synchrotron radiation facilities, and we would like to thank L. Chavas, P. Legrand, S. Sirigu, and P. Montaville for assistance in using beamline PROXIMA 1; G. Fox, M. Savko, and B. Shepard for assistance in using beamline PROXIMA 2A; and J. Perez and A. Thureau for assistance in using the beamline SWING. We thank the KAUST Bioscience Core Labs for their support, and we thank L. Jaremko for help with the NMR analysis. This research was supported by the Bill & Melinda Gates Foundation (OPP1136424); and the King Abdullah University of Science and Technology (KAUST).

### Author contributions
STA and SA-B conceived the idea for the study. USH and IH designed the experiments. STA collected and analyzed the SAXS data. STA and USH performed the protein structural studies. USH and IH performed the assays for characterization. XG collected and analyzed the NMR data. MJ and BAK did the *Striga* bioassays and analyzed the data. RAZ and DK helped in the protein studies. STA, SA-B, USH, and IH wrote the manuscript.

### Conflict of interest
The authors declare that they have no conflict of interest.

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
