## [Review Process File · EMBO Reports]

Structural basis for specific inhibition of the highly sensitive ShHTL7 receptor

Umar S. Hameed, Imran Haider, Muhammad Jamil, Boubacar A. Kountche, Xianrong Guo, Randa A. Zarban, Dongjin Kim, Salim Al-Babili, Stefan T. Arold

Review timeline:

Submission date:	9 December 2017
Editorial Decision:	25 January 2018
Revision received:	5 March 2018
Editorial Decision:	16 April 2018
Revision received:	6 May 2018
Editorial Decision:	6 June 2018
Revision received:	11 June 2018
Accepted:	19 June 2018

Transaction Report:

1st Editorial Decision

25 January 2018

Thank you for the submission of your manuscript to EMBO reports. I apologize for the delay in getting back to you; we have only recently received the final referee report, and all reports are copied below.

As you will see, while all referees acknowledge that the manuscript reports interesting observations, they also raise several important concerns that preclude publication of the manuscript in EMBO reports.

All three referees note that the efficiency of Triton X-100 in the seed germination assay is rather low and that micromolar concentrations are required to get a 50% inhibition. Moreover, the referees note that the reported nanomolar dissociation constant for Triton X-100 does not fit well with the slow association rate or the binding mode observed in the crystal structure. Importantly, referee 1 notes that the current data cannot exclude a more general effect of the detergent on seed germination.

Due to the nature of the criticisms, the amount of work likely to be required to address them, and the fact that EMBO reports can only invite revision of papers that receive enthusiastic support from the referees upon initial assessment, I am afraid that we do not feel it would be productive to call for a revised version of your manuscript at this stage.

Given the potential interest of your findings, we would, however, have no objections to consider a resubmission of the manuscript in the future if you were able to address all main concerns of the reviewers as highlighted above and in their reports. I think that it will be critical to provide further biochemical and physiological assays to unambiguously demonstrate that Triton X-100 specifically binds and inhibits ShHTL7 and *Striga* germination. I would like to stress though that such a manuscript would be treated as a new submission and would be evaluated again, also with respect to the literature and the novelty of your findings at the time of resubmission.

I apologize that I cannot be more positive at this point. I hope, however, that the referee comments are going to be helpful in strengthening your indeed very interesting initial observations and I will be happy to discuss any additional data on this topic with you in the future.

REFeree REPORTS

Referee #1:

The manuscript by Hameed et al., describes the identification of octyl phenol ethoxylate as a competitive inhibitor of the strigolactone receptor HTL-7 from the parasit plant *Striga hermonthica*. The authors set out to crystallize ShHTL-7 in the presence of the strigolactone analog GR24, but located a detergent molecule bound to the active site pocket of the receptor. The structural work is complemented with ligand binding and seed germination assays that together argue for Triton X 100 to act as an inhibitor of ShHTL7 function. Overall this is a potentially interesting piece of work of biotechnological relevance. In my opinion however, there are issues with the biochemical and physiological experiments, which need to be clarified and which I summarize below:

Major points:

- The authors in ref. 21 have reported an EC50 for ShHTL7 vs. GR24 in the picomolar range, supported by a seed germination assay (Fig. 2B in ref 21). How come that the apparent Kd of recombinant ShHTL7 vs. GR24 is only about 0.4 microM, as measured by Trp fluorescence (Fig. S4). The authors should clearly state this discrepancy, and provide a rationale for it. Fig. S4 should be part of main Fig. 3 (see below).
- Next, an apparent Kd for Triton X-100 is determined to be approx. 20 nM, by Triton, not by Trp fluorescence. Why did the authors choose to use different binding assays for the two molecules? Ideally, binding affinities for GR24 and Triton should be derived from the same method, and in this case it is essential to compare several methods. Structure-based point mutants that reduce binding of Triton and or GR24 should be shown alongside.
- I have difficulties rationalizing the nanomolar dissociation constant for Triton with the reported very slow association rate. Are the authors certain this is physiologically relevant and not an experimental artifact? Fig. 3A and S1c should be shown alongside. The figure legend in Fig. S1c seems to correspond to a different experiment. Also the 0.5% Triton used for the purification should be converted to molarities, so one can compare the Triton concentrations used for crystallization vs. the biochemical and physiological assays.
- In Fig. 3B, a change in protein aggregation temperature is estimated by static light scattering and used as a proxy for Triton X-100 binding to ShHTL7. Since Triton is a detergent which may reduce protein aggregation in a concentration dependent manner per se, is the change in aggregation temperature indeed related to ligand binding? A chemically distinct detergent with a similar CMC should be included as a control in this assay.
- Figs. 4a, 6a-g: I find the Yoshimulactone Green competition assay difficult to understand. First of all it would be good to know how much access YGL substrate is added to the reaction. To this end, molar enzyme concentrations should be provided, not micrograms. Next, addition of the YGL substrate should lead to an increase in fluorescence upon hydrolysis by all D14/HTL proteins, but no time course is shown in Fig. 4A or elsewhere. The inhibitory potential of Triton in this assay is thus hard to judge. In line with this, the estimated IC50 value is very different from the reported Kd for Triton, but no explanation is given. This should be carefully discussed, especially in light of the fact that the presented competition assay relies on a pre-incubation period.
- The GST pull-down shown in Fig. 4B lacks all required controls, such molecular weight markers, input controls, a mutant HTL7 that lacks Triton and or GR24 binding etc. This experiment has to be repeated with appropriate controls.

- Seed germination assay: As previously reported (compare ref. 21), nano- to picomolar concentrations of GR24 can trigger germination of striga seeds. As apparent from Figs. 4C and 6G, micromolar concentrations of Triton X-100 are required to even show a slight effect. This does not match the reported K_d values, and other effects may play a role here. For example it would be good to include a non-binding detergent with a CMC similar to Triton in these germination assays. Box-plots would be more informative compared to the presented bar diagrams, and the actual plates should be shown in the supplement.

Minor points:

- Fig. 2 could be move to the supplement.
- r.m.s.d.'s should be reported over a residue range (Fig. 3C, 5A, S3, S5, S6)
- structural superpositions should be shown as Ca traces rather than ribbon diagrams. (Fig. 3C, 5a)
- rather than predicting Triton binding based on homology modeling, actual binding data should be obtained, or this point should be omitted.

Referee #2:

In recent year, SLs have drawn much attention due to its important roles both in regulating shoot and root architectures of higher plants and in mediating interactions of host plants with symbiotic arbuscular mycorrhizal fungi and parasitic weeds. The witchweed has been a serious threat to agriculture in sub-Saharan Africa and successful Striga control is important and challenging. This research reported the structure of ShHTL7 and found that Triton X-100 specifically plugs the catalytic pocket of ShHTL7. Triton could repress SL binding and reduce the GR24-induced Striga germination, but the efficiency is relatively low.

The following are major points we concern.

1. As shown in Figure 1, there is only one hydrogen bond mediated by water in the crystal structure of ShHTL7-Triton, and this water molecular is pre-existing. The binding affinity should be relatively weak. Why the binding affinity is ten-fold stronger than GR24 in Fig. S4b?
2. As shown in Figure 4c, GR24 stimulated Striga germination at the concentration of 0.125-1 nM, while Triton X-100 showed about 50% reduction in GR24-induced germination at the concentration of 15.4 μ M. Thus, the efficiency of Triton X-100 in Striga-germination repression is low. This is inconsistent with the binding affinity of Triton and GR24.
3. Is there any conformational change between the ShHTL7-Apo (space group P65) and ShHTL7-Triton? If so, a structural superposition should be presented and RMSD values should be included in the manuscript. This information will be much more convincing than present data, including the superimposition between ShHTL7-Triton and modeled ShHTL7 bound to MAX2 in Fig S5b and the superimposition between the high R-factors Apo-ShHTL7(space group I222) and Triton-bound ShHTL7(space group P65) in Fig 3c.
4. Summary PDB validation reports should be submitted for all structures in Table 1, including ShHTL7-Apo(I222), ShHTL7-Triton, ShHTL7S95C-Triton, and ShHTL7-Apo(P65), to support whether the basic statistics (R-factor, number of reflections, resolution limits) are solid and no significant issue with model geometry is found.
5. Why are the R-factors so high in Table 1 (given the resolution exceeds 1.7Å for the ShHTL7-Apo-space group I222)? The refined structure should be inspected again, as the R-factors diverge from what one would expect at this resolution.
6. The presentation of manuscript could be improved to make it logic and well-organized.

There are also a few minor points:

1. A misspelling in Fig 1e, SHHTL8 should be ShHTL8?
2. The representation of Figure 4b is confusing. ShHTL7, ShHTL7-L143Y, and ShHTL7-H246A could be labeled directly in Figure 4b.

Referee #3:

In this manuscript the authors demonstrated that TritonX100 could bind to ShHTL7 by X-ray crystallography and inhibit SL-triggered germination of root parasitic weed, *Striga hermonthica*. Overall the manuscript is well written, logical and understandable. The data appears to be of high quality and repetitions are noted and stats are explained.

Followings are comments:

1. In Fig. 4 the authors stated that Triton does not affect rice phenotype, but they did not mention tillering, which is the most important phenotype induced by SL signal deficiency. In Fig. 4, it looks Triton treatment induces tillering in rice.
2. In in vitro assay, triton can bind ShHTL7 with high affinity but in the *Striga* germination assay the activity was not so high as expected from the in vitro assay.
3. By the oxidation of -SH to -SO₂H, molecular weight will increase by 32, but In Fig. 2, the MW of ShHTL7 increases 34. Is it correct?

1st Revision - authors' response

5 March 2018

Detailed reply to reviewer's comments

Referee #1:

The manuscript by Hameed et al., describes the identification of octyl phenol ethoxylate as a competitive inhibitor of the Strigolactone receptor HTL7 from the parasitic plant *Striga hermonthica*. The authors set out to crystallize ShHTL7 in the presence of the strigolactone analog GR24, but located a detergent molecule bound to the active site pocket of the receptor. The structural work is complemented with ligand binding and seed germination assays that together argue for Triton X 100 to act as an inhibitor of ShHTL7 function. Overall this is a potentially interesting piece of work of biotechnological relevance. In my opinion however, there are issues with the biochemical and physiological experiments, which need to be clarified and which I summarize below:

Major points:

- The authors in ref. 21 have reported an EC₅₀ for ShHTL7 vs. GR24 in the picomolar range, supported by a seed germination assay (Fig. 2B in ref 21). How come that the apparent K_d of recombinant ShHTL7 vs. GR24 is only about 0.4 microM, as measured by Trp fluorescence (Fig. S4). The authors should clearly state this discrepancy, and provide a rationale for it. Fig. S4 should be part of main Fig. 3 (see below).

- Response:

We thank the reviewer for raising this point. Actually, the discrepancy between in vitro and in planta measurements, which has been noticed by the reviewer, is already known from the literature. Toh et al., 2015 (ref. 22) reported that picomolar concentrations of GR24 are sufficient to trigger seed germination in transgenic *Arabidopsis* lines expressing *ShHTL7* and calculated an EC₅₀ (vs. GR24) of 17.5 picoM in this transgenic system. However, in vitro measurement of the activity of recombinant ShHTL7s revealed an IC₅₀ of ~0.5 μM (vs. GR24 in YLG assay) for ShHTL7 (Tsuchiya *et al.*, 2015, Science; ref. 21). Thus, our data are in line with the literature.

These studies show that in vitro binding/hydrolysis assays, GR24-based seed germination activity in transgenic *Arabidopsis*, and *Striga* germination through natural SLs in nature cannot be compared. The apparent SL sensitivity in planta may be strongly influenced by processes such as bioavailability, local SL concentration or interactions with other proteins. In the case of Triton, we assume that the inhibitory effect of this compound in planta is limited by its bioavailability and ability to be uptaken and transported within *Striga* seeds.

We have clarified these points in the amended manuscript (p4.; Section ‘Triton efficiently blocks binding of SLs to ShHLT7’ and Discussion).

- Fig. S4 should be part of main Fig. 3 (see below).

- Response:
Done.

- Next, an apparent K_d for Triton X-100 is determined to be approx. 20 nM, by Triton, not by Trp fluorescence. Why did the authors choose to use different binding assays for the two molecules? Ideally, binding affinities for GR24 and Triton should be derived from the same method, and in this case it is essential to compare several methods. Structure-based point mutants that reduce binding of Triton and or GR24 should be shown alongside.

- Response:
- We thank the reviewer for this notice. The usage of different methods was necessary due to the intrinsic Triton fluorescence that interferes with that of Trp. In addition, Triton does not reach the bottom of the active site upon binding, which is now illustrated by our new NMR data. Following the suggestion of the reviewer, we provide now data for both compounds by using the same method (microscale thermophoresis (MST)). These data (now **Figure 2d, e**) confirm that Triton binds significantly better than GR24. However, to avoid sticking of fluorescent molecules to the MST capillaries, we had to add a detergent (0.005 % of Tween-20), which may have an effect on the measured affinity for both hydrophobic compounds. This has been clearly noted in the text.

- I have difficulties rationalizing the nanomolar dissociation constant for Triton with the reported very slow association rate. Are the authors certain this is physiologically relevant and not an experimental artifact?

- Response:
We thank the reviewer for raising this point. The affinity of binding is defined as the ratio of the off-rate of the reaction to the on-rate of the reaction ($K_d = k_{off}/k_{on}$). Hence, a binding event with a slow on-rate can still be of a very high affinity if the off-rate is also extremely slow. This is exactly what happens. The slow on rate is explained by minor rearrangements in ShHTL7 that need to occur to allow binding. Once bound, triton release is extremely slow. Although we cannot say if the association rate is enhanced through additional factors *in vivo*, the time frame of this interaction is certainly sufficient for triggering *Striga* seed germination. This explanation is now reported in the text (p3; ‘ShHTL7 binds strongly and specifically to Triton *in vitro*’. *The fact that the Triton-ShHTL7 complex withstood size exclusion chromatography and extensive dialysis inferred that the off-rates of this association were extremely slow, resulting in the apparent high affinity despite the slow on-rate.*)

- Fig. 3A and S1c should be shown alongside. The figure legend in Fig. S1c seems to correspond to a different experiment. Also the 0.5% Triton used for the purification should be converted to molarities, so one can compare the Triton concentrations used for crystallization vs. the biochemical and physiological assays.

- Response:
Done

- In Fig. 3B, a change in protein aggregation temperature is estimated by static light scattering and used as a proxy for Triton X-100 binding to ShHTL7. Since Triton is a detergent which may reduce protein aggregation in a concentration dependent manner per se, is the change in aggregation temperature indeed related to ligand binding? A chemically distinct detergent with a similar CMC should be included as a control in this assay.

- Response:
We carried out an array of experiments to address this issue. We used the detergent n-Dodecyl Maltoside (nDM), which exhibits the same CMC as Triton but has different structure, as a control in *in vitro* and germination assays. In brief, *in vitro* we found that nDM, in contrast to Triton, does neither bind to ShHTL7 nor increase T_{agg} , nor block strigolactone binding/hydrolysis (using 3 different orthogonal assays: MST, YLG hydrolysis and DSLS). *In vivo*, we found that nDM does not block germination, while Triton does. Moreover, we also report *in vitro* and *in vivo* inhibitory action of PAA, which is not classified as a detergent. We have included these data and clarifying statements in the revised version.

- Figs. 4a, 6a-g: I find the Yoshimulactone Green competition assay difficult to understand. First of all it would be good to know how much excess YLG substrate is added to the reaction. To this end, molar enzyme concentrations should be provided, not micrograms.

- Response:
We thank the reviewer for this comment. For better understanding, we now stated that we used 3 μM of ShHTL7 and 1 μM of YLG. Since the product of the reaction remains bound to ShHTL7, preventing it from catalyzing further hydrolysis rounds, we had to use less YLG than ShHTL7.

Next, addition of the YLG substrate should lead to an increase in fluorescence upon hydrolysis by all D14/HTL proteins, but no time course is shown in Fig. 4A or elsewhere.

- Response:
We have added a time course now (Supplementary Figure 5c)

The inhibitory potential of Triton in this assay is thus hard to judge. In line with this, the estimated IC_{50} value is very different from the reported K_d for Triton, but no explanation is given. This should be carefully discussed, especially in light of the fact that the presented competition assay relies on a pre-incubation period.

- Response:
The fact that the IC_{50} value of Triton in this assay is not the same as the value for the affinity obtained by Triton fluorescence might be explained by the use of the hydrophobic organic solvent DMSO (0.1%) in the YLG assay, and by the fact that a IC_{50} value obtained in a competition assay is not the same as a directly measured K_d . This has now been discussed in the text (p4. 'Triton efficiently blocks binding of SLs to ShHTL7': *The observation that the IC_{50} value of Triton in this assay is similar to the K_d obtained by MST rather than to the affinity obtained by Triton fluorescence might be explained by the use of the hydrophobic organic solvent DMSO (0.1%) in the YLG assay, and by the fact that a IC_{50} value obtained in a competition assay is not the same as a directly measured K_d .)*

- The GST pull-down shown in Fig. 4B lacks all required controls, such molecular weight markers, input controls, a mutant HTL7 that lacks Triton and or GR24 binding etc. This experiment has to be repeated with appropriate controls.

- Response:
All the information requested was already in the Figure (now Fig. 4A), however we agree that it was difficult to match the previous figure legend with the data. We have now included more labels directly in the figure, showing clearly that there is an input control ('10% Input Control'), a mutant HTL7 that lacks Triton-binding capacity ('ShHTL7 L143Y mutant'), as well as a mutant that lacks GR24 binding capacity ('ShHTL7 H246N mutant'). We have amended the figure legend to include "The ShHTL7 mutations L143Y and H246N block binding to Triton and GR24, respectively. 10 % of used ShMAX2 are shown as input control." Additionally we have now included the full (uncut) WB and SDS page gels (with molecular weight markers) in the Supplementary material (Supplementary Figure 7a).

- Seed germination assay: As previously reported (compare ref. 21), nano- to picomolar concentrations of GR24 can trigger germination of striga seeds. As apparent from Figs. 4C and 6G, micromolar concentrations of Triton X-100 are required to even show a slight effect. This does not match the reported Kd values, and other effects may play a role here.

- Response:
Yes, we agree, as discussed above, and also in the revised version of the manuscript, this difference is interesting and suggests additional unknown factors.

For example it would be good to include a non-binding detergent with a CMC similar to Triton in these germination assays.

- Response:
Done – we used nDM as control (see our response above).

Box-plots would be more informative compared to the presented bar diagrams, and the actual plates should be shown in the supplement.

- Response:
Done; the plates are shown in Supplementary Figure 10.

Minor points:

- Fig. 2 could be moved to the supplement.

- Response:
We thank the reviewer for his suggestion. We have moved the figure accordingly.

- r.m.s.d.'s should be reported over a residue range (Fig. 3C, 5A, S3, S5, S6)

- Response:
Done.

- structural superpositions should be shown as Ca traces rather than ribbon diagrams. (Fig. 3C, 5a)

- Response:
Done.

- rather than predicting Triton binding based on homology modeling, actual binding data should be obtained, or this point should be omitted.

- Response:
Producing all 11 ShHTL7s and test binding would be beyond the scope of this manuscript. Based on our structural and (mutant) binding assays, we can predict with good confidence if a particular ShHTL sequence is likely to bind Triton or not, as the binding of this compound requires very specific pocket residues. We believe that this information, while not experimental, still adds value to the manuscript. However, we have clearly labelled it as a prediction in the manuscript. Of course we would take this part out of the manuscript if the editor and the reviewer agree that it would be necessary for acceptance of our paper.

Referee #2:

In recent year, SLs have drawn much attention due to its important roles both in regulating shoot

and root architectures of higher plants and in mediating interactions of host plants with symbiotic arbuscular mycorrhizal fungi and parasitic weeds. The witchweed has been a serious threat to agriculture in sub-Saharan Africa and successful *Striga* control is important and challenging. This research reported the structure of ShHTL7 and found that Triton X-100 specifically plugs the catalytic pocket of ShHTL7. Triton could repress SL binding and reduce the GR24-induced *Striga* germination, but the efficiency is relatively low.

The following are major points we concern.

1. As shown in Figure 1, there is only one hydrogen bond mediated by water in the crystal structure of ShHTL7-Triton, and this water molecular is pre-existing. The binding affinity should be relatively weak. Why the binding affinity is ten-fold stronger than GR24 in Fig. S4b?

- **Response:**

We thank the reviewer for raising this point. The biggest part of the binding strength comes from the hydrophobic interactions [in the text: The hydrocarbon moiety of Triton established hydrophobic interactions with Y26 (loop following β 2), L124, F134 (β 5- α 4 loop), M139, T142, L143, L146 (on α 4), F150 (α 4- α 5 loop), L153, T157, L161 (α 5), T190, I193, C194 (α 7), and M219 (β 6- α 8 loop) (Fig. 1c-e and Supplementary Fig. 2)]. The hydrogen bond between the Triton head moiety (i.e. the polyethylene oxide units) is not one of the pre-existing waters in the bottom of the binding pocket, but is captured by the triton ethylene oxide group at the outer rim of the binding pocket. This is now clarified in the text (p3; “The headgroup-ShHTL7 interactions with the outer rim of the ShHTL7 binding pocket comprised a water-mediated hydrogen bond (from the oxide of the first unit to the backbone nitrogen of E135)”).

2. As shown in Figure 4c, GR24 stimulated *Striga* germination at the concentration of 0.125-1 nM, while Triton X-100 showed about 50% reduction in GR24-induced germination at the concentration of 15.4 μ M. Thus, the efficiency of Triton X-100 in *Striga*-germination repression is low. This is inconsistent with the binding affinity of Triton and GR24.

- **Response:**

- We thank the reviewer for raising this point. Actually, the discrepancy between *in vitro* and *in planta* measurements, which has been noticed by the reviewer, is already known from the literature. Toh et al., 2015 (ref. 22) reported that picomolar concentrations of GR24 are sufficient to trigger seed germination in transgenic Arabidopsis lines expressing *ShHTL7* and calculated an EC_{50} (vs. GR24) of 17.5 picoM in this transgenic system. However, *in vitro* measurement of the activity of recombinant ShHTLs revealed an IC_{50} of \sim 0.5 μ M (vs. GR24 in YLG assay) for ShHTL7 (Tsuchiya et al., 2015, Science; ref. 21). Thus, our data are in line with the literature.
- These studies show that *in vitro* binding/hydrolysis assays, GR24-based seed germination activity in transgenic Arabidopsis, and *Striga* germination through natural SLs in nature cannot be compared. The apparent SL sensitivity *in planta* may be strongly influenced by processes such as bioavailability, local SL concentration or interactions with other proteins. In the case of Triton, we assume that the inhibitory effect of this compound *in planta* is limited by its bioavailability and ability to be uptaken and transported within *Striga* seeds. We have clarified these points in the amended manuscript (p4.; Section ‘Triton efficiently blocks binding of SLs to ShHLLT7’ and Discussion).

3. Is there any conformational change between the ShHTL7-Apo (space group P65) and ShHTL7-Triton? If so, a structural superposition should be presented and RMSD values should be included in the manuscript. This information will be much more convincing than present data, including the superimposition between ShHTL7-Triton and modeled ShHTL7 bound to MAX2 in Fig S5b and the superimposition between the high R-factors Apo-ShHTL7(space group I222) and Triton-bound ShHTL7(space group P65) in Fig 3c.

- **Response:**

In the former manuscript, we had presented a superimposition of the Triton-bound form in space group P65 and the apo form in I222, because the crystal packing in P65 precludes free movement of the helix $\alpha 4$. Indeed, the apo and Triton bound forms are almost identical. Nonetheless, we have now included this figure in the Supplementary Figures (the text p4 now reads: Triton-free ShHTL7 structures obtained in the same space group (P65) as the Triton-bound structures did not exhibit significant changes in the binding site (Supplementary Figure 5d).

To further address the amplitude of changes occurring upon Triton binding, we have now added a study using solution NMR and SAXS, which show that the changes introduced upon Triton binding are minor compared to the ones occurring following GR24 hydrolysis (Figure 3a and Supplementary Figure S4c, d). These data further support the suggested mechanism for GR24-ShHTL7:MAX2 interactions.

4. Summary PDB validation reports should be submitted for all structures in Table 1, including ShHTL7-Apo(I222), ShHTL7-Triton, ShHTL7S95C-Triton, and ShHTL7-Apo(P65), to support whether the basic statistics (R-factor, number of reflections, resolution limits) are solid and no significant issue with model geometry is found.

Response:

We have pdb codes for all structures after validation by PDB database, they are 5z82 – WT (I222 space group), 5z89 – S95C mutant bound to Triton X-100, 5z8p – WT (P65 space group), 5z95 – WT bound to Triton. These are now provided.

5. Why are the R-factors so high in Table 1 (given the resolution exceeds 1.7Å for the ShHTL7-Apo-space group I222)? The refined structure should be inspected again, as the R-factors diverge from what one would expect at this resolution.

Response:

The Phenix Polygon evaluation indicates that the values are within those expected for the resolution. We have nonetheless used an additional refinement algorithm, leading to the slightly improved R and Rfree values of 19.41 and 22.05, respectively, which are within the expected values for this resolution. No obvious discrepancies of the model from the data could be seen in the FoFc maps. I suspect that the increased dynamics of the helices 4 and 5 (which cannot be modelled very well) in this apo crystal form might lead to somewhat higher-than-expected R factors.

6. The presentation of manuscript could be improved to make it logic and well-organized.

Response:

We thank the reviewer for this suggestion. We have now added section headings and reorganized and reformulated several sections. We hope that this has improved the clarity of the manuscript.

There are also a few minor points:

1. A misspelling in Fig 1e, SHHTL8 should be ShHTL8?

Response:

Done.

2. The representation of Figure 4b is confusing. ShHTL7, ShHTL7-L143Y, and ShHTL7-H246A could be labeled directly in Figure 4b.

Response:

The legend has been rewritten for clarity, and the molecules have been labelled directly in the figure, as suggested, now Fig 4a.

Referee #3:

In this manuscript the authors demonstrated that TritonX100 could bind to ShHTL7 by X-ray crystallography and inhibit SL-triggered germination of root parasitic weed, *Striga hermonthica*. Overall the manuscript is well written, logical and understandable. The data appears to be of high quality and repetitions are noted and stats are explained.

Followings are comments:

1. In Fig. 4 the authors stated that Triton does not affect rice phenotype, but they did not mention tillering, which is the most important phenotype induced by SL signal deficiency. In Fig. 4, it looks Triton treatment induces tillering in rice.

Response:

We thank the reviewer for this comment. Actually, we did not perform a tillering study because our *in vitro* assays exclude binding of Triton by rice D14. In addition, we did not see significant differences in tillering upon Triton treatment in Fig. 4d.

2. In *in vitro* assay, triton can bind ShHTL7 with high affinity but in the *Striga* germination assay the activity was not so high as expected from the *in vitro* assay.

Response:

We thank the reviewer for raising this point. Indeed, it appears that *in vitro* binding/hydrolysis assays and seed germination activity *in planta* cannot be compared with each other (Please also see our reply to reviewers 1 and 2 on this topic). For instance, Toh et al., 2015 (ref. 21) reported that GR24 EC50, calculated from seed germination activity of *htl-3* Arabidopsis line expressing ShHTL7, is around 17.5 pM. However, *in vitro* measurement of the activity of recombinant ShHTLs revealed an IC50 of ~0.5 μ M (vs. GR24 in YLG assay) for ShHTL7 (Tsuchiya et al., 2015, Science; ref. 20). In case of Triton, we assume that the inhibitory effect of this compound *in planta* is also determined by its bioavailability and ability to be uptaken and transported within *Striga* seeds. We are currently designing new Triton-based compounds that can be more efficient in inhibiting *Striga* seed germination.

We have clarified these points in the amended manuscript (p4.; Section ‘Triton efficiently blocks binding of SLs to ShHTL7’ and Discussion: “Interestingly, the picomolar sensitivity of ShHTLs *in planta* does not correlate with their affinity for SLs *in vitro*, inferring that other factors contribute^{21,22}. The discrepancy between *in vitro* and *in planta* measurements is in line with previous observations. Toh et al. reported that picomolar concentrations of GR24 are sufficient to trigger seed germination in transgenic Arabidopsis lines expressing *ShHTL7*, whereas their *in vitro* YLG assay with recombinant ShHTL6 and ShHTL7 revealed an IC_{50} of ~0.5 μ M for GR24^{21,22}. Thus, *in vitro* binding or hydrolysis assays, GR24-based seed germination activity in transgenic Arabidopsis, and *Striga* germination through natural SLs in the field cannot be compared. The apparent SL sensitivity in *planta* may be strongly influenced by processes such as bioavailability, local SL concentration or

interactions with other proteins. Hence, the inhibitory effect of Triton *in planta* might be limited by its bioavailability and ability to be uptaken and transported within *Striga* seeds”).

3. By the oxidation of -SH to -SO₂H, molecular weight will increase by 32, but In Fig. 2, the MW of ShHTL7 increases 34. Is it correct?

Response

Thank you for noting this. It's only a typo in the figure and has been changed now.

2nd Editorial Decision

16 April 2018

Thank you for the submission of your revised manuscript to EMBO reports. I apologize again for my delayed response but as you will see from the reports copied below, the referee opinions on your manuscript remain divided.

As you will see, referee 2 and 3 support publication in EMBO reports, while referee 1 does not. It appears that there is one important concern that remains valid also after the revision and this concern is shared by referee 1 and 2: the nanomolar affinity for Triton X-100 calculated from the triton fluorescent experiments does not match the micromolar affinity seen with microscale thermophoresis experiments. It is also not in good agreement with the identical 15N HSQC spectra for the apo and Triton-bound form of HTL7. It appears that the seed germination assays are also not supportive of high affinity binding between Triton and ShHTL7 since Triton is not very effective in inhibition GR24-induced *Striga* germination.

I have discussed this concern further with the referees. Referee 1 concluded that it remains difficult to judge from the current data if Triton X-100 can indeed act as receptor agonist. Referee 3 pointed out that the nanomolar affinity for Triton X-100 calculated from the Triton fluorescent experiments could also result from an unspecific binding of Triton to the surface of ShHTL. To address this question, referee 3 suggested to calculate the nanomolar affinity for Triton X-100 to mutant ShHTL7 to which Triton cannot bind, such as the L143Y and H246N mutants. This would help to discriminate the contribution of specific binding to the binding pocket from unspecific binding to the receptor surface.

Given the persisting concerns regarding the specific action of Triton X-100 as ShHTL7 agonist, which are very much at the heart of the current paper, we cannot offer publication of your manuscript in EMBO reports at this point. Having said so, we also recognize that two referees support publication of your findings in EMBO reports. We have therefore decided to give you the exceptional possibility of another round of revision to address these concerns.

I look forward to seeing a revised form of your manuscript when it is ready.

REFeree REPORTS

Referee #1:

The revised manuscript by Hameed et al. characterizes the detergent Triton-X-100 as a competitive inhibitor of the *Striga* HTL receptors required for germination. Many of the points raised in my initial review have been experimentally addressed by the authors, but the new data do, in my opinion, not strengthen the main claim of this paper, namely that Triton X-100 is a potent competitive inhibitor of ShHTL7. I provide a point-by-point response below:

Initial review:

- Next, an apparent K_d for Triton X-100 is determined to be approx. 20 nM, by Triton, not by Trp fluorescence. Why did the authors choose to use different binding assays for the two molecules? Ideally, binding affinities for GR24 and Triton should be derived from the same method, and in this case it is essential to compare several methods. Structure-based point mutants that reduce binding of Triton and or GR24 should be shown alongside.

-

-

Response:

We thank the reviewer for this notice. The usage of different methods was necessary due to the intrinsic Triton fluorescence that interferes with that of Trp. In addition, Triton does not reach the bottom of the active site upon binding, which is now illustrated by our new NMR data. Following the suggestion of the reviewer, we provide now data for both compounds by using the same method (microscale thermophoresis (MST)). These data (now Figure 2d, e) confirm that Triton binds significantly better than GR24. However, to avoid sticking of fluorescent molecules to the MST capillaries, we had to add a detergent (0.005 % of Tween-20), which may have an effect on the measured affinity for both hydrophobic compounds. This has been clearly noted in the text.

My response:

The K_d calculated from the triton fluorescence experiments is ~20 nM, from the microscale thermophoresis experiment is 0.44 micromolar (~ 20 fold lower). These differences could indeed be method dependent, but why are the apo and Triton bound ^{15}N HSQC spectra in Fig. 3 so similar? For a specific binding event with low to high nanomolar affinity, one would at least expect ten peaks or so to shift significantly, but the spectra look virtually identical! Given the fact that also the ITC experiment was inconclusive, there are now two binding assays presented in the manuscript that indicate no binding of Triton X-100 to HTL7 (NMR and ITC, could that be due to the cmc of Triton??) and two assays which suggest either very high affinity binding, or binding in the same range as observed for GR24. Based on this, I find it an overstatement that "ShHTL7 bound Triton with a markedly stronger affinity than GR24".

Initial review:

- I have difficulties rationalizing the nanomolar dissociation constant for Triton with the reported very slow association rate. Are the authors certain this is physiologically relevant and not an experimental artifact?

Response:

We thank the reviewer for raising this point. The affinity of binding is defined as the ratio of the off-rate of the reaction to the on-rate of the reaction ($K_d = k_{off}/k_{on}$). Hence, a binding event with a slow on-rate can still be of a very high affinity if the off-rate is also extremely slow. This is exactly what happens. The slow on rate is explained by minor rearrangements in ShHTL7 that need to occur to allow binding. Once bound, triton release is extremely slow. Although we cannot say if the association rate is enhanced through additional factors in vivo, the time frame of this interaction is certainly sufficient for triggering Striga seed germination. This explanation is now reported in the text (p3; 'ShHTL7 binds strongly and specifically to Triton in vitro'. The fact that the Triton-ShHTL7 complex withstood size exclusion chromatography and extensive dialysis inferred that the off-rates of this association were extremely slow, resulting in the apparent high affinity despite the slow on-rate.)

My response:

Given a K_d of 20 nM and an association rate in the range of hours the dissociation rate would have to be so slow that no protein half life known to me could match that. So either the association rate is different, or, somewhat more likely, the K_d is much lower than estimated by Triton fluorescence.

Referee #2:

The revised manuscript has addressed our most concerns, but some concerns need to be addressed:

1. As shown in Figure 4b and Figure S10, GR24 stimulated Striga germination at the concentration

of 0.125-1 nM, while Triton X-100 showed about 50% reduction in GR24-induced germination at the concentration of 15.4 μ M. Thus, the efficiency of Triton X-100 in Striga-germination repression is low. This is inconsistent with the *in vitro* binding affinity of Triton and GR24. Is that due to the absorbing problem?

2. We suggest the summary PDB validation reports should be submitted for all structures in Table 1, including ShHTL7-Apo (I222), ShHTL7- Triton, ShHTL7S95C- Triton, and ShHTL7- Apo(P65), to support whether the basic statistics (R-factor, number of reflections, resolution limits) are solid and no significant issue with model geometry is found.

3. As shown in Figure 4d and Figure S5a, the MST measurement k_d value is 0.4 μ m, why the ITC measurement has not a bit cal. change?

Referee #3:

I recommend this ms to be published in EMBOR.

2nd Revision - authors' response

6 May 2018

Point-by-point reply to reviewers

Referee #1:

The revised manuscript by Hameed et al. characterizes the detergent Triton-X-100 as a competitive inhibitor of the Striga HTL receptors required for germination. Many of the points raised in my initial review have been experimentally addressed by the authors, but the new data do, in my opinion, not strengthen the main claim of this paper, namely that Triton X-100 is a potent competitive inhibitor of ShHTL7. I provide a point-by-point response below:

Initial review:

Next, an apparent K_d for Triton X-100 is determined to be approx. 20 nM, by Triton, not by Trp fluorescence. Why did the authors choose to use different binding assays for the two molecules? Ideally, binding affinities for GR24 and Triton should be derived from the same method, and in this case it is essential to compare several methods. Structure-based point mutants that reduce binding of Triton and or GR24 should be shown alongside.

Response:

We thank the reviewer for this notice. The usage of different methods was necessary due to the intrinsic Triton fluorescence that interferes with that of Trp. In addition, Triton does not reach the bottom of the active site upon binding, which is now illustrated by our new NMR data. Following the suggestion of the reviewer, we provide now data for both compounds by using the same method (microscale thermophoresis (MST)). These data (now Figure 2d, e) confirm that Triton binds significantly better than GR24. However, to avoid sticking of fluorescent molecules to the MST capillaries, we had to add a detergent (0.005 % of Tween-20), which may have an effect on the measured affinity for both hydrophobic compounds. This has been clearly noted in the text.

My response:

The K_d calculated from the triton fluorescence experiments is \sim 20 nM, from the microscale thermophoresis experiment is 0.44 micromolar (\sim 20 fold lower).

AUTHOR REPLY:

We thank this reviewer for insisting on this discrepancy between experimental methods (MST and Triton fluorescence). We initially thought that this difference was a result of the (required) presence of detergents in some, but not all methods. We have therefore repeated Triton fluorescence experiments multiple times with several replicates in buffer with or without detergent (tween-20). All the results converge on an affinity of \sim 250 nM, which is now consistent with the affinity

observed in MST. The best explanation for the difference with previous fluorescence measurements is that there was a mistake in the protein concentration in the previous manuscript versions. We are glad that this issue is solved, and we have updated text and figures.

These differences could indeed be method dependent, but why are the apo and Triton bound ¹⁵N HSQC spectra in Fig. 3 so similar? For a specific binding event with low to high nanomolar affinity, one would at least expect ten peaks or so to shift significantly, but the spectra look virtually identical!

AUTHOR REPLY:

In our initial version, the small size and choice of color in this subfigure might have made it difficult to see the differences between Triton-bound and free states in the ¹⁵N HSQC spectra. We have now prepared the same figure with different colors and also made a separate bigger supplementary figure (Supplementary Fig.7) to highlight the changes induced by triton binding in HSQC experiments. We only highlighted changes that are from well-dispersed chemical shifts and did not include changes that might also be explained by changes in noise level of the data. We discussed these data additionally with another NMR spectroscopist who was not involved in the study to further exclude any bias.

We hope that the reviewer now agrees that the number and nature of chemical shifts in the ¹⁵N HSQC between Triton-bound and free states are in the expected range (Supplementary Fig.7). We would also like to highlight that the Triton-bound sample was co-purified with Triton (no additional Triton was added during the NMR experiment), and the observed Triton has persisted despite gel-filtration chromatography and extensive dialysis with buffer without Triton.

Another important aspect of our NMR experiments is to show that Triton effectively blocks the structural deformation induced by GR24 on ShHTL7 in absence of Triton. This experiment provides a strong evidence that Triton acts as competitive and specific inhibitor.

Given the fact that also the ITC experiment was inconclusive, there are now two binding assays presented in the manuscript that indicate no binding of Triton X-100 to HTL7 (NMR and ITC, could that be due to the cmc of Triton??) and two assays which suggest either very high affinity binding, or binding in the same range as observed for GR24. Based on this, I find it an overstatement that "ShHTL7 bound Triton with a markedly stronger affinity than GR24".

AUTHOR REPLY:

We showed that the ShHTL7-Triton association only reaches its maximum after 2.5h, probably because the association is limited by the available population of ShHTL7 with helix α 4 in the open conformation. However, ITC measures the heat changes within 300 seconds or less following injection of the ligand. The much slower reaction dynamics explain why we do not see any significant heat change in ITC, despite the strong binding. In fact, what ITC is 'seeing' would correspond to a Triton fluorescence binding curve between the blue (0 minutes) and red (30 minutes) binding curve of Figure 2a.

Conversely, we now show strong binding under identical and different buffer conditions in MST, fluorescence Triton fluorescence. Moreover, strong binding is necessary for the Triton-ShHTL7 complex to persist gel-filtration chromatography (for which the K_d typically needs to be below 300 nM) and extensive dialysis. We show that the binding is specific, and that crystallography-derived protein mutants or other detergents do not bind. Additional evidence for strong binding comes from the NMR analysis (chemical shift changes, and blocking of GR24 effects), YLG assays, and the DARTS proteolysis assay (now added to the new version, Supplementary Fig. 5c). Moreover, we also show that the Triton-derived (but non-detergent) compound PAA also binds, excluding non-specific detergent effects.

Give all this abundant evidence for direct and specific binding, we have decided to take out the ITC experiment, which appears to be misleading. However, in light of the new data, we have changed the sentence "ShHTL7 bound Triton with a markedly stronger affinity than GR24". In to: "ShHTL7 bound Triton with a slightly stronger affinity than GR24". We note that the binding difference of Triton compared to GR24 remains significant (in terms of error margin).

Initial review:

I have difficulties rationalizing the nanomolar dissociation constant for Triton with the reported very slow association rate. Are the authors certain this is physiologically relevant and not an experimental artifact?

Response:

We thank the reviewer for raising this point. The affinity of binding is defined as the ratio of the off-rate of the reaction to the on-rate of the reaction ($K_d = k_{off}/k_{on}$). Hence, a binding event with a slow on-rate can still be of a very high affinity if the off-rate is also extremely slow. This is exactly what happens. The slow on rate is explained by minor rearrangements in ShHTL7 that need to occur to allow binding. Once bound, triton release is extremely slow. Although we cannot say if the association rate is enhanced through additional factors *in vivo*, the time frame of this interaction is certainly sufficient for triggering Striga seed germination. This explanation is now reported in the text (p3; 'ShHTL7 binds strongly and specifically to Triton *in vitro*'. The fact that the Triton-ShHTL7 complex withstood size exclusion chromatography and extensive dialysis inferred that the off-rates of this association were extremely slow, resulting in the apparent high affinity despite the slow on-rate.)

My response:

Given a K_d of 20 nM and an association rate in the range of hours the dissociation rate would have to be so slow that no protein half life known to me could match that. So either the association rate is different, or, somewhat more likely, the K_d is much lower than estimated by Triton fluorescence.

AUTHOR REPLY:

As outlined above, and documented in the new manuscript version, our results now consistently give a K_d of ~250 nM across methods and buffers for the Triton-ShHTL7 association. As noted in the new manuscript version, the affinity of the association cannot be much lower, since it persists even throughout dialysis and gel filtration. The slow k_{on} might rather reflect the low probability for Triton to encounter ShHTL7 molecules where $\alpha 4$ is in the required open conformation. Hence, the apparent slow k_{on} may reflect the dynamics of ShHTL7 $\alpha 4$.

Referee #2:

The revised manuscript has addressed our most concerns, but some concerns need to be addressed:

1. As shown in Figure 4b and Figure S10, GR24 stimulated Striga germination at the concentration of 0.125-1 nM, while Triton X-100 showed about 50% reduction in GR24-induced germination at the concentration of 15.4 μ M. Thus, the efficiency of Triton X-100 in Striga-germination repression is low. This is inconsistent with the *in vitro* binding affinity of Triton and GR24. Is that due to the absorbing problem?

AUTHOR REPLY:

Yes, currently the *in vivo* efficiency of Triton is still low compared to what could be expected from its *in vitro* affinity (however it is the only specific inhibitor known!). Hence, the inhibitory effect of Triton in planta appears to be limited by its bioavailability and ability to be taken up and transported within Striga seeds (this statement is found at the end of the 4th paragraph of the Discussion section). There is indeed a marked discrepancy between *in vivo* efficacy and *in vitro* affinity also for GR24, as already observed previously by other groups. Understanding these mechanisms would surely help improving the efficacy of Triton or other herbicides targeting Striga seeds.

2. We suggest the summary PDB validation reports should be submitted for all structures in Table 1, including ShHTL7-Apo (I222), ShHTL7- Triton, ShHTL7S95C- Triton, and ShHTL7- Apo(P65), to support whether the basic statistics (R-factor, number of reflections, resolution limits) are solid and no significant issue with model geometry is found.

AUTHOR REPLY:

Thank you for this suggestions. These reports are included in the revised version.

3. As shown in Figure 4d and Figure S5a, the MST measurement k_d value is 0.4 μ m, why the ITC measurement has not a bit cal. change?

AUTHOR REPLY:

We showed that the Triton-ShHTL7 association only reaches its maximum after 2.5h, probably because the association is limited by the available population of ShHTL7 with helix $\alpha 4$ in the open conformation. However, ITC measures the heat changes within 300 seconds or less following injection of the ligand. The much slower reaction dynamics explain why we do not see any significant heat change in ITC, despite the strong binding. In fact, what ITC is 'seeing' would correspond to a Triton fluorescence binding curve between the blue (0 minutes) and red (30 minutes) binding curve of Figure 2a. In other words, we could not observe heat changes in ITC because the binding time scale is not adapted for this method.

Given that the ITC data appear to be misleading, and that we have a wealth of data showing strong binding, we have decided to take out the ITC experiment.

Referee #3:

I recommend this ms to be published in EMBOR.

3rd Editorial Decision

6 June 2018

Thank you for the submission of your further revised manuscript to our journal. It has been sent back to former referee 1, who now also supports publication of the manuscript in EMBO reports.

Browsing through the manuscript myself, I noticed several things that we need before we can proceed with the formal acceptance of your study.

- The Appendix needs a title page with a table of content including page numbers. Please resubmit the Appendix as a pdf file instead of a word file.

- You can promote up to 5 Appendix figures to the Expanded View format. EV figures will be displayed in the main HTML of the paper in a collapsible format. If you wish to do so, please upload these figures as individual .tif files and include their legends in a separate section called Expanded View Legends after the main Figure Legends section. Otherwise, they can remain part of the Appendix.

- Please provide up to five keywords.

- Please specify the contribution of Xianrong Guo in the Author contribution section.

- Please change the title of the Reference section to "References" and please re-format the reference list. Currently many references list only the first author followed by et al. The correct format lists the first ten authors followed by et al.

You can download the respective EndNote file from our online Guide to Authors
<https://drive.google.com/file/d/0BxFM9n2lEE5oOHM4d2xEbmpxN2c/view>

- It appears that Figure 5d-f is described in the text after Figures 6 and 7. Could you please check this and if so, you might consider a rearrangement of the figures since their order should follow their callouts in the text.

- Please replace the callouts to the Supplementary figures in the text. You have correctly labeled these figures as Appendix Figure Sx in the Appendix file but you refer to them as Supplementary Fig. x in the text. This also applies to Appendix table S1, which is referred to as table S1 in the text.

- We noticed that some of the figures get "fuzzy" when the image is magnified by 200 % (e.g., Fig. 2a, b or i). Could you please submit the tif files at higher resolution, if possible?

- You have deposited the structural data in PDB. Please provide the accession codes in a "Data Availability" section at the end of Materials & Methods (suggested wording: "The [protein interaction | microarray | mass spectrometry] data from this publication have been deposited to the [name of the database] database [URL] and assigned the identifier [accession | permalink | hashtag]")."

- Figure 6a-e: please define the number of experiments and the nature of the error bars in the figure legend.

- Appendix figure S5a, S6a: please define the number of experiments and the nature of the error bars in the legend.

- Moreover, I made some changes/suggestions to the Abstract (see attached file).

- Finally, EMBO reports papers are accompanied online by A) a short (1-2 sentences) summary of the findings and their significance, B) 2-3 bullet points highlighting key results and C) a synopsis image that is 550x200-400 pixels large (width x height). You can either show a model or key data in the synopsis image. Please note that the size is rather small and that text needs to be readable at the final size. Please send us this information along with the revised manuscript.

REFEREE REPORT

Referee #1:

The new data are now consistent and support binding of the detergent to the receptor. I think it would be fair now to move this paper to the press.

3rd Revision - authors' response

11 June 2018

The authors made all suggested changes.

Corresponding Author Name: Stefan T. Arold, Salim Al-Babili

Manuscript Number: EMBOR-2017-45619V2